# THE ROLE OF OVER-PARAMETRIZATION IN GENERALIZATION OF NEURAL NETWORKS

**Behnam Neyshabur**
School of Mathematics
Institute for Advanced Study
Princeton, NJ 08540
bneyshabur@gmail.com

**Zhiyuan Li**
Department of Computer Science
Princeton University
Princeton, NJ 08540
zhiyuanli@princeton.edu

**Srinadh Bhojanapalli**
Toyota Technological
Institute at Chicago
Chicago, IL 60637
srinadh@ttic.edu

**Yann LeCun**
Department of Computer Science
New York University New York, NY 10012
yann@cs.nyu.edu

**Nathan Srebro**
Toyota Technological
Institute at Chicago Chicago, IL 60637
nati@ttic.edu

## ABSTRACT

Despite existing work on ensuring generalization of neural networks in terms of scale sensitive complexity measures, such as norms, margin and sharpness, these complexity measures do not offer an explanation of why neural networks generalize better with over-parametrization. In this work we suggest a novel complexity measure based on unit-wise capacities resulting in a *tighter* generalization bound for two layer ReLU networks. Our capacity bound correlates with the behavior of test error with increasing network sizes (within the range reported in the experiments), and could partly explain the improvement in generalization with over-parametrization. We further present a matching lower bound for the Rademacher complexity that improves over previous capacity lower bounds for neural networks.

## 1 INTRODUCTION

Deep neural networks have enjoyed great success in learning across a wide variety of tasks. They played a crucial role in the seminal work of Krizhevsky et al. (2012), starting an arms race of training larger networks with more hidden units, in pursuit of better test performance (He et al., 2016). In fact the networks used in practice are over-parametrized to the extent that they can easily fit random labels to the data (Zhang et al., 2017). Even though they have such a high capacity, when trained with real labels they achieve smaller generalization error.

Traditional wisdom in learning suggests that using models with increasing capacity will result in overfitting to the training data. Hence capacity of the models is generally controlled either by limiting the size of the model (number of parameters) or by adding an explicit regularization, to prevent from overfitting to the training data. Surprisingly, in the case of neural networks we notice that increasing the model size only helps in improving the generalization error, even when the networks are trained without any explicit regularization - weight decay or early stopping (Lawrence et al., 1998; Srivastava et al., 2014; Neyshabur et al., 2015c). In particular, Neyshabur et al. (2015c) observed that training on models with increasing number of hidden units lead to decrease in the test error for image classification on MNIST and CIFAR-10. Similar empirical observations have been made over a wide range of architectural and hyper-parameter choices (Liang et al., 2017; Novak et al., 2018; Lee et al., 2018). What explains this improvement in generalization with over-parametrization? What is the right measure of complexity of neural networks that captures this generalization phenomenon?

Complexity measures that depend on the total number of parameters of the network, such as VC bounds, do not capture this behavior as they increase with the size of the network. Existing works suggested different norm, margin and sharpness based measures, to measure the capacity of neural networks, in an attempt to explain the generalization behavior observed in practice (Neyshabur et al., 2015b; Keskar et al., 2017; Dziugaite & Roy, 2017; Neyshabur et al., 2017; Bartlett et al., 2017;

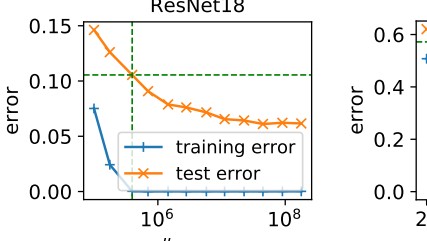 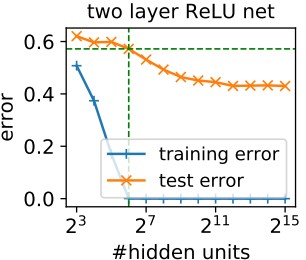 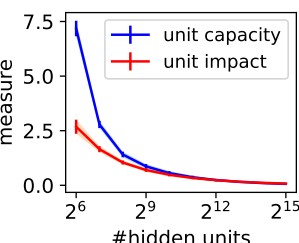

Figure 1: Over-parametrization phenomenon. **Left panel**: Training pre-activation ResNet18 architecture of different sizes on CIFAR-10 dataset. We observe that even when after network is large enough to completely fit the training data(reference line), the test error continues to decrease for larger networks. **Middle panel**: Training fully connected feedforward network with single hidden layer on CIFAR-10. We observe the same phenomena as the one observed in ResNet18 architecture. **Right panel**: Unit capacity captures the complexity of a hidden unit and unit impact captures the impact of a hidden unit on the output of the network, and are important factors in our capacity bound (Theorem 1). We observe empirically that both average unit capacity and average unit impact shrink with a rate faster than $1/\sqrt{h}$ where $h$ is the number of hidden units. Please see Supplementary Section A for experiments settings.

Neyshabur et al., 2018; Golowich et al., 2018; Arora et al., 2018). In particular, Bartlett et al. (2017) showed a margin based generalization bound that depends on the spectral norm and $\ell_{1,2}$ norm of the layers of a network. However, as shown in Neyshabur et al. (2017) and in Figure 5, these complexity measures fail to explain why over-parametrization helps, and in fact increase with the size of the network. Dziugaite & Roy (2017) numerically evaluated a generalization bound based on PAC-Bayes. Their reported numerical generalization bounds also increase with the increasing network size. These existing complexity measures increase with the size of the network, even for two layer networks, as they depend on the number of hidden units either explicitly, or the norms in their measures implicitly depend on the number of hidden units for the networks used in practice (Neyshabur et al., 2017) (see Figures 3 and 5).

To study and analyze this phenomenon more carefully, we need to simplify the architecture making sure that the property of interest is preserved after the simplification. We therefore chose two layer ReLU networks since as shown in the left and middle panel of Figure 1, it exhibits the same behavior with over-parametrization as the more complex pre-activation ResNet18 architecture. In this paper we prove a *tighter* generalization bound (Theorem 2) for two layer ReLU networks. Our capacity bound, unlike existing bounds, correlates with the test error and decreases with the increasing number of hidden units, in the experimental range considered. Our key insight is to characterize complexity at a unit level, and as we see in the right panel in Figure 1 these unit level measures shrink at a rate faster than $1/\sqrt{h}$ for each hidden unit, decreasing the overall measure as the network size increases. When measured in terms of layer norms, our generalization bound depends on the Frobenius norm of the top layer and the Frobenius norm of the difference of the hidden layer weights with the initialization, which decreases with increasing network size (see Figure 2).

The closeness of learned weights to initialization in the over-parametrized setting can be understood by considering the limiting case as the number of hidden units go to infinity, as considered in Bengio et al. (2006) and Bach (2017). In this extreme setting, just training the top layer of the network, which is a convex optimization problem for convex losses, will result in minimizing the training error, as the randomly initialized hidden layer has all possible features. Intuitively, the large number of hidden units here represent all possible features and hence the optimization problem involves just picking the right features that will minimize the training loss. This suggests that as we over-parametrize the networks, the optimization algorithms need to do less work in tuning the weights of the hidden units to find the right solution. Dziugaite & Roy (2017) indeed have numerically evaluated a PAC-Bayes measure from the initialization used by the algorithms and state that the Euclidean distance to the initialization is smaller than the Frobenius norm of the parameters. Nagarajan & Kolter (2017) also make a similar empirical observation on the significant role of initialization, and in fact prove an initialization dependent generalization bound for linear networks. However they do not prove a similar generalization bound for neural networks. Alternatively, Liang et al. (2017) suggested a

Fisher-Rao metric based complexity measure that correlates with generalization behavior in larger networks, but they also prove the capacity bound only for *linear* networks.

**Contributions:** Our contributions in this paper are as follows.

- We empirically investigate the role of over-parametrization in generalization of neural networks on 3 different datasets (MNIST, CIFAR10 and SVHN), and show that the existing complexity measures increase with the number of hidden units - hence do not explain the generalization behavior with over-parametrization.

- We prove *tighter* generalization bounds (Theorem 2) for two layer ReLU networks, improving over previous results. Our proposed complexity measure for neural networks decreases with the increasing number of hidden units, in the experimental range considered (see Section 2), and can potentially explain the effect of over-parametrization on generalization of neural networks.

- We provide a matching lower bound for the Rademacher complexity of two layer ReLU networks with a scalar output. Our lower bound considerably improves over the best known bound given in Bartlett et al. (2017), and to our knowledge is the first such lower bound that is bigger than the Lipschitz constant of the network class.

## 1.1 PRELIMINARIES

We consider two layer fully connected ReLU networks with input dimension $d$, output dimension $c$, and the number of hidden units $h$. Output of a network is $f_{\mathbf{V},\mathbf{U}}(\mathbf{x}) = \mathbf{V}[\mathbf{U}\mathbf{x}]_+$[1] where $\mathbf{x} \in \mathbb{R}^d$, $\mathbf{U} \in \mathbb{R}^{h \times d}$ and $\mathbf{V} \in \mathbb{R}^{c \times h}$. We denote the incoming weights to the hidden unit $i$ by $\mathbf{u}_i$ and the outgoing weights from hidden unit $i$ by $\mathbf{v}_i$. Therefore $\mathbf{u}_i$ corresponds to row $i$ of matrix $\mathbf{U}$ and $\mathbf{v}_i$ corresponds to the column $i$ of matrix $\mathbf{V}$.

We consider the $c$-class classification task where the label with maximum output score will be selected as the prediction. Following Bartlett et al. (2017), we define the margin operator $\mu : \mathbb{R}^c \times [c] \to \mathbb{R}$ as a function that given the scores $f(\mathbf{x}) \in \mathbb{R}^c$ for each label and the correct label $y \in [c]$, it returns the difference between the score of the correct label and the maximum score among other labels, i.e. $\mu(f(\mathbf{x}), y) = f(\mathbf{x})[y] - \max_{i \neq y} f(\mathbf{x})[i]$. We now define the ramp loss as follows:

$$\ell_\gamma(f(\mathbf{x}), y) = \begin{cases} 0 & \mu(f(\mathbf{x}), y) > \gamma \\ \mu(f(\mathbf{x}), y)/\gamma & \mu(f(\mathbf{x}), y) \in [0, \gamma] \\ 1 & \mu(f(\mathbf{x}), y) < 0. \end{cases} \tag{1}$$

For any distribution $\mathcal{D}$ and margin $\gamma > 0$, we define the expected margin loss of a predictor $f(.)$ as $L_\gamma(f) = \mathbb{P}_{(\mathbf{x},y) \sim \mathcal{D}}[\ell_\gamma(f(\mathbf{x}), y)]$. The loss $L_\gamma(.)$ defined this way is bounded between 0 and 1. We use $\hat{L}_\gamma(f)$ to denote the empirical estimate of the above expected margin loss. As setting $\gamma = 0$ reduces the above to classification loss, we will use $L_0(f)$ and $\hat{L}_0(f)$ to refer to the expected risk and the training error respectively.

## 2 GENERALIZATION OF TWO LAYER RELU NETWORKS

For any function class $\mathcal{F}$, let $\ell_\gamma \circ \mathcal{F}$ denote the function class corresponding to the composition of the loss function and functions from class $\mathcal{F}$. With probability $1 - \delta$ over the choice of the training set of size $m$, the following generalization bound holds for any function $f \in \mathcal{F}$ (Mohri et al., 2012, Theorem 3.1):

$$L_0(f) \leq \hat{L}_\gamma(f) + 2\mathcal{R}_S(\ell_\gamma \circ \mathcal{F}) + 3\sqrt{\frac{\ln(2/\delta)}{2m}}. \tag{2}$$

where $\mathcal{R}_S(\mathcal{H})$ is the Rademacher complexity of a class $\mathcal{H}$ of functions with respect to the training set $\mathcal{S}$ which is defined as:

$$\mathcal{R}_{\mathcal{S}}(\mathcal{H}) = \frac{1}{m} \mathbb{E}_{\xi \sim \{\pm 1\}^m} \left[ \sup_{f \in \mathcal{H}} \sum_{i=1}^m \xi_i f(x_i) \right]. \tag{3}$$

---

[1]Since the number of bias parameters is negligible compare to the size of the network, we drop the bias parameters to simplify the analysis. Moreover, one can model the bias parameters in the first layer by adding an extra dimension with value 1.

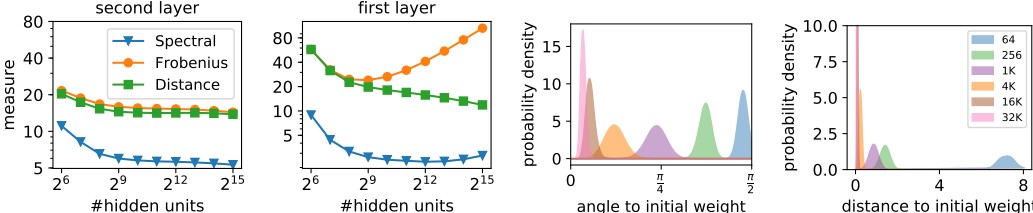

Figure 2: Properties of two layer ReLU networks trained on CIFAR-10. We report different measures on the trained network. From left to right: measures on the second (output) layer, measures on the first (hidden) layer, distribution of angles of the trained weights to the initial weights in the first layer, and the distribution of unit capacities of the first layer. "Distance" in the first two plots is the distance from initialization in Frobenius norm.

Rademacher complexity is a capacity measure that captures the ability of functions in a function class to fit random labels which increases with the complexity of the class.

## 2.1 AN EMPIRICAL INVESTIGATION

We will bound the Rademacher complexity of neural networks to get a bound on the generalization error. Since the Rademacher complexity depends on the function class considered, we need to choose the right function class that only captures the real trained networks, which is potentially much smaller than networks with all possible weights, to get a complexity measure that explains the decrease in generalization error with increasing width. Choosing a bigger function class can result in weaker capacity bounds that do not capture this phenomenon. Towards that we first investigate the behavior of different measures of network layers with increasing number of hidden units. The experiments discussed below are done on the CIFAR-10 dataset. Please see Section A for similar observations on SVHN and MNIST datasets.

**First layer:** As we see in the second panel in Figure 2 even though the spectral and Frobenius norms of the learned layer decrease initially, they eventually increase with $h$, with Frobenius norm increasing at a faster rate. However the distance Frobenius norm, measured w.r.t. initialization ($\|\mathbf{U} - \mathbf{U}_0\|_F$), decreases. This suggests that the increase in the Frobenius norm of the weights in larger networks is due to the increase in the Frobenius norm of the random initialization. To understand this behavior in more detail we also plot the distance to initialization per unit and the distribution of angles between learned weights and initial weights in the last two panels of Figure 2. We indeed observe that per unit distance to initialization decreases with increasing $h$, and a significant shift in the distribution of angles to initial points, from being almost orthogonal in small networks to almost aligned in large networks. This per unit distance to initialization is a key quantity that appears in our capacity bounds and we refer to it as unit capacity in the remainder of the paper.

*Unit capacity.* We define $\beta_i = \left\|\mathbf{u}_i - \mathbf{u}_i^0\right\|_2$ as the unit capacity of the hidden unit $i$.

**Second layer:** Similar to first layer, we look at the behavior of different measures of the second layer of the trained networks with increasing $h$ in the first panel of Figure 2. Here, unlike the first layer, we notice that Frobenius norm and distance to initialization both decrease and are quite close suggesting a limited role of initialization for this layer. Moreover, as the size grows, since the Frobenius norm $\|\mathbf{V}\|_F$ of the second layer slightly decreases, we can argue that the norm of outgoing weights $\mathbf{v}_i$ from a hidden unit $i$ decreases with a rate faster than $1/\sqrt{h}$. If we think of each hidden unit as a linear separator and the top layer as an ensemble over classifiers, this means the impact of each classifier on the final decision is shrinking with a rate faster than $1/\sqrt{h}$. This per unit measure again plays an important role and we define it as unit impact for the remainder of this paper.

*Unit impact.* We define $\alpha_i = \|\mathbf{v}_i\|_2$ as the unit impact, which is the magnitude of the outgoing weights from the unit $i$.

Motivated by our empirical observations we consider the following class of two layer neural networks that depend on the capacity and impact of the hidden units of a network. Let $\mathcal{W}$ be the following restricted set of parameters:

$$\mathcal{W} = \left\{ (\mathbf{V}, \mathbf{U}) \mid \mathbf{V} \in \mathbb{R}^{c \times h}, \mathbf{U} \in \mathbb{R}^{h \times d}, \|\mathbf{v}_i\| \leq \alpha_i, \left\|\mathbf{u}_i - \mathbf{u}_i^0\right\|_2 \leq \beta_i \right\}, \tag{4}$$

We now consider the hypothesis class of neural networks represented using parameters in the set $\mathcal{W}$:

$$\mathcal{F}_{\mathcal{W}} = \left\{ f(\mathbf{x}) = \mathbf{V}\left[\mathbf{U}\mathbf{x}\right]_+ \mid (\mathbf{V}, \mathbf{U}) \in \mathcal{W} \right\}. \tag{5}$$

Our empirical observations indicate that networks we learn from real data have bounded unit capacity and unit impact and therefore studying the generalization behavior of the above function class can potentially provide us with a better understanding of these networks. Given the above function class, we will now study its generalization properties.

## 2.2 GENERALIZATION BOUND

In this section we prove a generalization bound for two layer ReLU networks. We first bound the Rademacher complexity of the class $\mathcal{F}_{\mathcal{W}}$ in terms of the sum over hidden units of the product of unit capacity and unit impact. Combining this with the equation (2) will give us the generalization bound.

**Theorem 1.** *Given a training set $\mathcal{S} = \{\mathbf{x}_i\}_{i=1}^m$ and $\gamma > 0$, Rademacher complexity of the composition of loss function $\ell_\gamma$ over the class $\mathcal{F}_{\mathcal{W}}$ defined in equations (4) and (5) is bounded as follows:*

$$\mathcal{R}_{\mathcal{S}}(\ell_\gamma \circ \mathcal{F}_{\mathcal{W}}) \leq \frac{2\sqrt{2c}}{\gamma m} \sum_{j=1}^h \alpha_j \left( \beta_j \|\mathbf{X}\|_F + \|\mathbf{u}_j^0 \mathbf{X}\|_2 \right) \tag{6}$$

$$\leq \frac{2\sqrt{2c}}{\gamma\sqrt{m}} \|\boldsymbol{\alpha}\|_2 \left( \|\boldsymbol{\beta}\|_2 \sqrt{\frac{1}{m}\sum_{i=1}^m \|\mathbf{x}_i\|_2^2} + \sqrt{\frac{1}{m}\sum_{i=1}^m \|\mathbf{U}^0 \mathbf{x}_i\|_2^2} \right). \tag{7}$$

The proof is given in the supplementary Section C. The main idea behind the proof is a new technique to decompose the complexity of the network into complexity of the hidden units. To our knowledge, all previous works decompose the complexity to that of layers and use Lipschitz property of the network to bound the generalization error. However, Lipschitzness of the layer is a rather weak property that ignores the linear structure of each individual layer. Instead, by decomposing the complexity across the hidden units, we get the above tighter bound on the Rademacher complexity of the two layer neural networks.

The generalization bound in Theorem 1 is for any function in the function class defined by a specific choice of $\boldsymbol{\alpha}$ and $\boldsymbol{\beta}$ fixed before the training procedure. To get a generalization bound that holds for all networks, it suffices to cover the space of possible values for $\boldsymbol{\alpha}$ and $\boldsymbol{\beta}$ and take a union bound over it. The following theorem states the generalization bound for any two layer ReLU network [2].

**Theorem 2.** *For any $h \geq 2$, $\gamma > 0$, $\delta \in (0,1)$ and $\mathbf{U}^0 \in \mathbb{R}^{h \times d}$, with probability $1 - \delta$ over the choice of the training set $\mathcal{S} = \{\mathbf{x}_i\}_{i=1}^m \subset \mathbb{R}^d$, for any function $f(\mathbf{x}) = \mathbf{V}[\mathbf{U}\mathbf{x}]_+$ such that $\mathbf{V} \in \mathbb{R}^{c \times h}$ and $\mathbf{U} \in \mathbb{R}^{h \times d}$, the generalization error is bounded as follows:*

$$L_0(f) \leq \hat{L}_\gamma(f) + \tilde{O}\left( \frac{\sqrt{c}\|\mathbf{V}\|_F \left( \|\mathbf{U} - \mathbf{U}^0\|_F \|\mathbf{X}\|_F + \|\mathbf{U}^0 \mathbf{X}\|_F \right)}{\gamma m} + \sqrt{\frac{h}{m}} \right)$$

$$\leq \hat{L}_\gamma(f) + \tilde{O}\left( \frac{\sqrt{c}\|\mathbf{V}\|_F \left( \|\mathbf{U} - \mathbf{U}^0\|_F + \|\mathbf{U}^0\|_2 \right) \sqrt{\frac{1}{m}\sum_{i=1}^m \|\mathbf{x}_i\|_2^2}}{\gamma\sqrt{m}} + \sqrt{\frac{h}{m}} \right).$$

The above generalization bound empirically improves over the existing bounds, and decreases with increasing width for networks learned in practice (see Section 2.3). We also show an explicit lower bound for the Rademacher complexity (Theorem 3), matching the first term in the above generalization bound, thereby showing its tightness. The additive factor $\tilde{O}(\sqrt{h/m})$ in the above bound is the result of taking the union bound over the cover of $\boldsymbol{\alpha}$ and $\boldsymbol{\beta}$. As we see in Figure 5, in the regimes of interest this additive term is small and does not dominate the first term, resulting in an overall decrease in capacity with over-parametrization. In Appendix Section B, we further extend the generalization bound in Theorem 2 to $\ell_p$ norms, presenting a finer tradeoff between the two terms.

## 2.3 COMPARISON WITH EXISTING RESULTS

In table 1 we compare our result with the existing generalization bounds, presented for the simpler setting of two layer networks. In comparison with the bound

---

[2]For the statement with exact constants see Lemma 14 in Supplementary Section C.

| # | Reference | Measure |
|---|-----------|---------|
| (1) | Harvey et al. (2017) | $\tilde{\Theta}(dh)$ |
| (2) | Bartlett & Mendelson (2002) | $\tilde{\Theta}\left(\|\mathbf{U}\|_{\infty,1}\|\mathbf{V}\|_{\infty,1}\right)$ |
| (3) | Neyshabur et al. (2015b), Golowich et al. (2018) | $\tilde{\Theta}\left(\|\mathbf{U}\|_F\|\mathbf{V}\|_F\right)$ |
| (4) | Bartlett et al. (2017), Golowich et al. (2018) | $\tilde{\Theta}\left(\|\mathbf{U}\|_2\|\mathbf{V}-\mathbf{V}_0\|_{1,2}+\|\mathbf{U}-\mathbf{U}_0\|_{1,2}\|\mathbf{V}\|_2\right)$ |
| (5) | Neyshabur et al. (2018) | $\tilde{\Theta}\left(\|\mathbf{U}\|_2\|\mathbf{V}-\mathbf{V}_0\|_F+\sqrt{h}\|\mathbf{U}-\mathbf{U}_0\|_F\|\mathbf{V}\|_2\right)$ |
| (6) | Theorem 2 | $\tilde{\Theta}\left(\|\mathbf{U}_0\|_2\|\mathbf{V}\|_F+\left\|\mathbf{U}-\mathbf{U}^0\right\|_F\|\mathbf{V}\|_F+\sqrt{h}\right)$ |

Table 1: Comparison with the existing generalization measures presented for the case of two layer ReLU networks with constant number of outputs and constant margin.

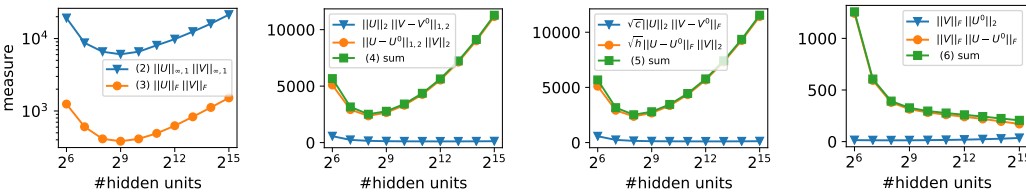

Figure 3: Behavior of terms presented in Table 1 with respect to the size of the network trained on CIFAR-10.

$\tilde{\Theta}\left(\|\mathbf{U}\|_2\|\mathbf{V}-\mathbf{V}_0\|_{1,2}+\|\mathbf{U}-\mathbf{U}_0\|_{1,2}\|\mathbf{V}\|_2\right)$ (Bartlett et al., 2017; Golowich et al., 2018): The first term in their bound $\|\mathbf{U}\|_2\|\mathbf{V}-\mathbf{V}_0\|_{1,2}$ is of smaller magnitude and behaves roughly similar to the first term in our bound $\|\mathbf{U}_0\|_2\|\mathbf{V}\|_F$ (see Figure 3 last two panels). The key complexity term in their bound is $\|\mathbf{U}-\mathbf{U}_0\|_{1,2}\|\mathbf{V}\|_2$, and in our bound is $\left\|\mathbf{U}-\mathbf{U}^0\right\|_F\|\mathbf{V}\|_F$, for the range of $h$ considered. $\|\mathbf{V}\|_2$ and $\|\mathbf{V}\|_F$ differ by number of classes, a small constant, and hence behave similarly. However, $\|\mathbf{U}-\mathbf{U}_0\|_{1,2}$ can be as big as $\sqrt{h}\cdot\left\|\mathbf{U}-\mathbf{U}^0\right\|_F$ when most hidden units have similar capacity. Infact their bound increases with $h$ mainly because of this term $\|\mathbf{U}-\mathbf{U}_0\|_{1,2}$. As we see in the first and second panels of Figure 3, $\ell_1$ norm terms appearing in Bartlett & Mendelson (2002); Bartlett et al. (2017); Golowich et al. (2018) over hidden units increase with the number of units as the hidden layers learned in practice are usually dense. Neyshabur et al. (2015b); Golowich et al. (2018) showed a bound depending on the product of Frobenius norms of layers, which increases with $h$, showing the important role of initialization in our bounds. In fact the proof technique of Neyshabur et al. (2015b) does not allow for getting a bound with norms measured from initialization, and our new decomposition approach is the key for the tighter bound.

**Experimental comparison.** We train two layer ReLU networks of size $h$ on CIFAR-10 and SVHN datasets with values of $h$ ranging from $2^6$ to $2^{15}$. The training and test error for CIFAR-10 are shown in the first panel of Figure 1, and for SVHN in the left panel of Figure 4. We observe for both datasets that even though a network of size 128 is enough to get to zero training error, networks with sizes well beyond 128 can still get better generalization, even when trained without any regularization. We further measure the unit-wise properties introduce in the paper, namely unit capacity and unit impact. These quantities decrease with increasing $h$, and are reported in the right panel of Figure 1 and second panel of Figure 4. Also notice that the number of epochs required for each network size to get 0.01 cross-entropy loss decreases for larger networks as shown in the third panel of Figure 4.

For the same experimental setup, Figure 5 compares the behavior of different capacity bounds over networks of increasing sizes. Generalization bounds typically scale as $\sqrt{C/m}$ where $C$ is the effective capacity of the function class. The left panel reports the effective capacity $C$ based on different measures calculated with all the terms and constants. We can see that our bound is the only that decreases with $h$ and is consistently lower that other norm-based data-independent bounds. Our bound even improves over VC-dimension for networks with size larger than 1024. While the actual numerical values are very loose, we believe they are useful tools to understand the relative generalization behavior with respect to different complexity measures, and in many cases applying a set of data-dependent techniques, one can improve the numerical values of these bounds significantly (Dziugaite & Roy, 2017; Arora et al., 2018). In the middle and right panel we presented

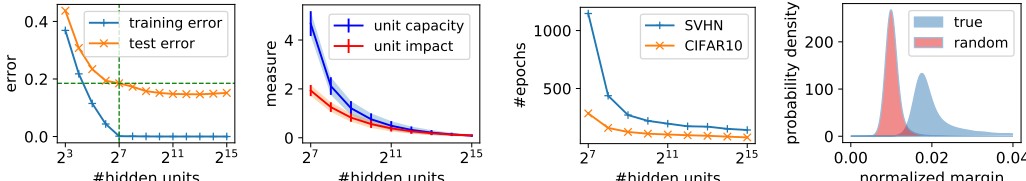

Figure 4: First panel: Training and test errors of fully connected networks trained on SVHN. Second panel: unit-wise properties measured on a two layer network trained on SVHN dataset. Third panel: number of epochs required to get 0.01 cross-entropy loss. Fourth panel: comparing the distribution of margin of data points normalized on networks trained on true labels vs a network trained on random labels.

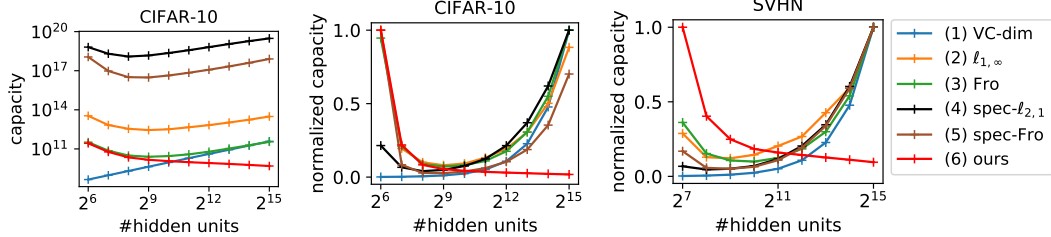

Figure 5: Left panel: Comparing network capacity bounds on CIFAR10 (unnormalized). Middle panel: Comparing capacity bounds on CIFAR10 (normalized). Right panel: Comparing capacity bounds on SVHN (normalized).

each capacity bound normalized by its maximum in the range of the study for networks trained on CIFAR-10 and SVHN respectively. For both datasets, our capacity bound is the only one that decreases with the size even for networks with about 100 million parameters. All other existing norm-based bounds initially decrease for smaller networks but then increase significantly for larger networks. Our capacity bound therefore could potentially point to the right properties that allow the over-parametrized networks to generalize.

Finally we check the behavior of our complexity measure under a different setting where we compare this measure between networks trained on real and random labels (Neyshabur et al., 2017; Bartlett et al., 2017). We plot the distribution of margin normalized by our measure, computed on networks trained with true and random labels in the last panel of Figure 4 - and as expected they correlate well with the generalization behavior.

## 3  LOWER BOUND

In this section we will prove a lower bound for the Rademacher complexity of neural networks, that matches the dominant term in the upper bound of Theorem 1. We will show our lower bound on a smaller function class than $\mathcal{F}_{\mathcal{W}}$, with an additional constraint on spectral norm of the hidden layer. This allows for comparison with the existing results, and also extends the lower bound to the bigger class $\mathcal{F}_{\mathcal{W}}$.

**Theorem 3.** *Define the parameter set*

$$\mathcal{W}' = \left\{ (\mathbf{V}, \mathbf{U}) \mid \mathbf{V} \in \mathbb{R}^{1 \times h}, \mathbf{U} \in \mathbb{R}^{h \times d}, \|\mathbf{v}_j\| \le \alpha_j, \left\|\mathbf{u}_j - \mathbf{u}_j^0\right\|_2 \le \beta_j, \|\mathbf{U} - \mathbf{U}^0\|_2 \le \max_{j \in h} \beta_j \right\},$$

*and let $\mathcal{F}_{\mathcal{W}'}$ be the function class defined on $\mathcal{W}'$ by equation (5). Then, for any $d = h \le m$, $\{\alpha_j, \beta_j\}_{j=1}^h \subset \mathbb{R}^+$ and $\mathbf{U}_0 = \mathbf{0}$, there exists $\mathcal{S} = \{\mathbf{x}_i\}_{i=1}^m \subset \mathbb{R}^d$, such that*

$$\mathcal{R}_{\mathcal{S}}(\mathcal{F}_{\mathcal{W}}) \ge \mathcal{R}_{\mathcal{S}}(\mathcal{F}_{\mathcal{W}'}) = \Omega \left( \frac{\sum_{j=1}^h \alpha_j \beta_j \|\mathbf{X}\|_F}{m} \right).$$

Clearly, $\mathcal{W}' \subseteq \mathcal{W}$, since it has an extra constraint. The complete proof is given in the supplementary Section C.3.

The above complexity lower bound matches the first term, $\frac{\sum_{i=1}^{h} \alpha_i \beta_i \|\mathbf{X}\|_F}{m\gamma}$, in the upper bound of Theorem 1, upto $\frac{1}{\gamma}$, which comes from the $\frac{1}{\gamma}$-Lipschitz constant of the ramp loss $l_\gamma$.

To match the second term in the upper bound for Theorem 1, consider the setting with $c = 1$ and $\boldsymbol{\beta} = \mathbf{0}$, resulting in,

$$\mathcal{R}_\mathcal{S}(\mathcal{F}_\mathcal{W}) = \mathcal{R}_{[\mathbf{U}_0 \circ \mathcal{S}]_+}(\mathcal{F}_\mathcal{V}) = \sum_{j=1}^{h} \Omega\left(\frac{\alpha_j \|\mathbf{u}_j^0 \mathbf{X}\|_2}{m}\right) = \Omega\left(\frac{\sum_{j=1}^{h} \alpha_j \|\mathbf{u}_j^0 \mathbf{X}\|_2}{m}\right),$$

where $\mathcal{F}_\mathcal{V} = \{f(\mathbf{x}) = \mathbf{V}\mathbf{x} \mid \mathbf{V} \in \mathbb{R}^{1 \times h}, \|\mathbf{v}_j\| \leq \alpha_j\}$. In other words, when $\boldsymbol{\beta} = \mathbf{0}$, the function class $\mathcal{F}_{\mathcal{W}'}$ on $\mathcal{S} = \{\mathbf{x}_i\}_{i=1}^m$ is equivalent to the linear function class $\mathcal{F}_\mathcal{V}$ on $[\mathbf{U}_0 \circ \mathcal{S}]_+ = \{[\mathbf{U}_0\mathbf{x}_i]_+\}_{i=1}^m$, and therefore we have the above lower bound, showing that the upper bound provided in Theorem 1 is tight. It also indicates that even if we have more information, such as bounded spectral norm with respect to the reference matrix is small (which effectively bounds the Lipschitz of the network), we still cannot improve our upper bound.

To our knowledge, all the previous capacity lower bounds for spectral norm bounded classes of neural networks with a scalar output and element-wise activation functions correspond to the Lipschitz constant of the network. Our lower bound strictly improves over this, and shows a gap between the Lipschitz constant of the network (which can be achieved by even linear models), and the capacity of neural networks. This lower bound is non-trivial, in the sense that the smaller function class excludes the neural networks with all rank-1 matrices as weights, and thus shows a $\Theta(\sqrt{h})$-capacity gap between the neural networks with ReLU activations and linear networks. The lower bound therefore does not hold for linear networks. Finally, one can extend the construction in this bound to more layers by setting all the weight matrices in the intermediate layers to be the Identity matrix.

**Comparison with existing results.** Bartlett et al. (2017) have proved a Rademacher complexity lower bound of $\Omega\left(\frac{s_1 s_2 \|\mathbf{X}\|_F}{m}\right)$ for the function class defined by the parameter set:

$$\mathcal{W}_{\text{spec}} = \left\{(\mathbf{V}, \mathbf{U}) \mid \mathbf{V} \in \mathbb{R}^{1 \times h}, \mathbf{U} \in \mathbb{R}^{h \times d}, \|\mathbf{V}\|_2 \leq s_1, \|\mathbf{U}\|_2 \leq s_2\right\}. \tag{8}$$

Note that $s_1 s_2$ is the Lipschitz bound of the function class $\mathcal{F}_{\mathcal{W}_{spec}}$. Given $\mathcal{W}_{spec}$ with bounds $s_1$ and $s_2$, choosing $\boldsymbol{\alpha}$ and $\boldsymbol{\beta}$ such that $\|\boldsymbol{\alpha}\|_2 = s_1$ and $\max_{i \in [h]} \beta_i = s_2$ results in $\mathcal{W}' \subset \mathcal{W}_{spec}$. Hence we get the following result from Theorem 3, showing a stronger lower bound for this function class as well.

**Corollary 4.** $\forall h = d \leq m$, $s_1, s_2 \geq 0$, $\exists \mathcal{S} \in \mathbb{R}^{d \times m}$ such that $\mathcal{R}_\mathcal{S}(\mathcal{F}_{\mathcal{W}_{spec}}) = \Omega\left(\frac{s_1 s_2 \sqrt{h} \|\mathbf{X}\|_F}{m}\right)$.

Hence our result improves the lower bound in Bartlett et al. (2017) by a factor of $\sqrt{h}$. Theorem 7 in Golowich et al. (2018) also gives a $\Omega(s_1 s_2 \sqrt{c})$ lower bound, $c$ is the number of outputs of the network, for the composition of 1-Lipschitz loss function and neural networks with bounded spectral norm, or $\infty$-Schatten norm. Our above result even improves on this lower bound.

# 4 DISCUSSION

In this paper we present a new capacity bound for neural networks that empirically decreases with the increasing number of hidden units, and could potentially explain the better generalization performance of larger networks. In particular, we focused on understanding the role of width in the generalization behavior of two layer networks. More generally, understanding the role of depth and the interplay between depth and width in controlling capacity of networks, remain interesting directions for future study. We also provided a matching lower bound for the capacity improving on the current lower bounds for neural networks. While these bounds are useful for relative comparison between networks of different size, their absolute values still remain larger than the number of training samples, and it is of interest to get bounds with numerically smaller values.

In this paper we do not address the question of whether optimization algorithms converge to low complexity networks in the function class considered in this paper, or in general how does different hyper parameter choices affect the complexity of the recovered solutions. It is interesting to understand the implicit regularization effects of the optimization algorithms (Neyshabur et al., 2015a; Gunasekar et al., 2017; Soudry et al., 2018) for neural networks, which we leave for future work.

## ACKNOWLEDGEMENTS

The authors thank Sanjeev Arora for many fruitful discussions on generalization of neural networks and David McAllester for discussion on the distance to random initialization. The authors also thank Wei Zhan for discussion on the lower bound. This research was supported in part by NSF IIS-RI award 1302662 and Schmidt Foundation.

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

## A    EXPERIMENTS

### A.1    EXPERIMENTS SETTINGS

Below we describe the setting for each reported experiment.

**ResNet18**    In this experiment, we trained a pre-activation ResNet18 architecture on CIFAR-10 dataset. The architecture consists of a convolution layer followed by 8 residual blocks (each of which consist of two convolution) and a linear layer on the top. Let $k$ be the number of channels in the first convolution layer. The number of output channels and strides in residual blocks is then $[k, k, 2k, 2k, 4k, 4k, 8k, 8k]$ and $[1, 1, 1, 2, 1, 2, 1, 2]$ respectively. Finally, we use the kernel sizes 3 in all convolutional layers. We train 11 architectures where for architecture $i$ we set $k = \lceil 2^{2+i/2} \rceil$. In each experiment we train using SGD with mini-batch size 64, momentum 0.9 and initial learning rate 0.1 where we reduce the learning rate to 0.01 when the cross-entropy loss reaches 0.01 and stop when the loss reaches 0.001 or if the number of epochs reaches 1000. We use the reference line in the plots to differentiate the architectures that achieved 0.001 loss. We do not use weight decay or dropout but perform data augmentation by random horizontal flip of the image and random crop of size $28 \times 28$ followed by zero padding.

**Two Layer ReLU Networks**    We trained fully connected feedforward networks on CIFAR-10, SVHN and MNIST datasets. For each data set, we trained 13 architectures with sizes from $2^3$ to $2^{15}$ each time increasing the number of hidden units by factor 2. For each experiment, we trained the network using SGD with mini-batch size 64, momentum 0.9 and fixed step size 0.01 for MNIST and 0.001 for CIFAR-10 and SVHN. We did not use weight decay, dropout or batch normalization. For experiment, we stopped the training when the cross-entropy reached 0.01 or when the number of epochs reached 1000. We use the reference line in the plots to differentiate the architectures that achieved 0.01 loss.

**Evaluations**    For each generalization bound, we have calculated the exact bound including the log-terms and constants. We set the margin to 5th percentile of the margin of data points. Since bounds in Bartlett & Mendelson (2002) and Neyshabur et al. (2015c) are given for binary classification, we multiplied Bartlett & Mendelson (2002) by factor $c$ and Neyshabur et al. (2015c) by factor $\sqrt{c}$ to make sure that the bound increases linearly with the number of classes (assuming that all output units have the same norm). Furthermore, since the reference matrices can be used in the bounds given in Bartlett et al. (2017) and Neyshabur et al. (2018), we used random initialization as the reference matrix. When plotting distributions, we estimate the distribution using standard Gaussian kernel density estimation.

### A.2    SUPPLEMENTARY FIGURES

Figures 6 and 7 show the behavior of several measures on networks with different sizes trained on SVHN and MNIST datasets respectively. The left panel of Figure 8 shows the over-parametrization phenomenon in MNSIT dataset and the middle and right panels compare our generalization bound to others.

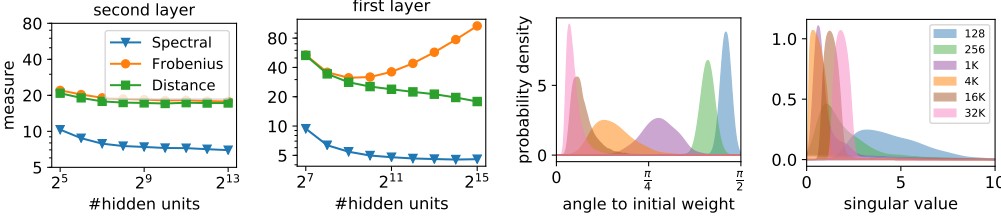

Figure 6: Different measures on fully connected networks with a single hidden layer trained on SVHN. From left to right: measure on the output layer, measures in the first layer, distribution of angle to initial weight in the first layer, and singular values of the first layer.

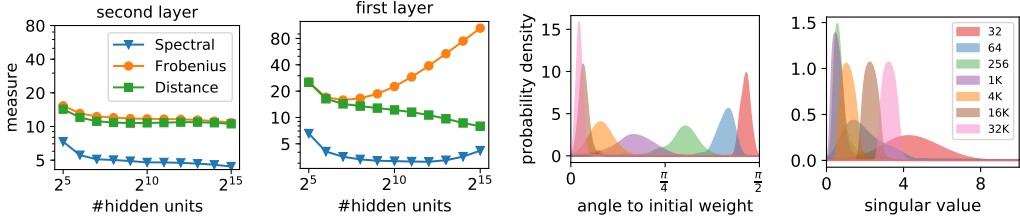

Figure 7: Different measures on fully connected networks with a single hidden layer trained on MNIST. From left to right: measure on the output layer, measures in the first layer, distribution of angle to initial weight in the first layer, and singular values of the first layer.

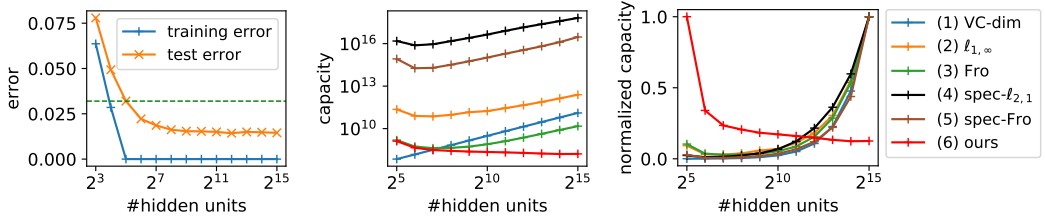

Figure 8: Left panel: Training and test errors of fully connected networks trained on MNIST. Middle panel: Comparing capacity bounds on MNIST (normalized). Left panel: Comparing capacity bounds on MNIST (unnormalized).

## B  EXTENDING THE GENERALIZATION BOUND TO $\ell_p$ NORM

In this section we generalize the Theorem 2 to $\ell_p$ norm. The main new ingredient in the proof is the Lemma 11, in which we construct a cover for the $\ell_p$ ball with entry-wise dominance.

**Theorem 5.** *For any $h, p \geq 2$, $\gamma > 0$, $\delta \in (0, 1)$ and $\mathbf{U}^0 \in \mathbb{R}^{h \times d}$, with probability $1 - \delta$ over the choice of the training set $\mathcal{S} = \{\mathbf{x}_i\}_{i=1}^m \subset \mathbb{R}^d$, for any function $f(\mathbf{x}) = \mathbf{V}[\mathbf{U}\mathbf{x}]_+$ such that $\mathbf{V} \in \mathbb{R}^{c \times h}$ and $\mathbf{U} \in \mathbb{R}^{h \times d}$, the generalization error is bounded as follows:*

$$L_0(f) \leq \hat{L}_\gamma(f) + \tilde{O}\left( \frac{\sqrt{c}h^{\frac{1}{2} - \frac{1}{p}} \left\| \mathbf{V}^T \right\|_{p,2} \left( h^{\frac{1}{2} - \frac{1}{p}} \left\| \mathbf{U} - \mathbf{U}^0 \right\|_{p,2} \left\| \mathbf{X} \right\|_F + \left\| \mathbf{U}^0\mathbf{X} \right\|_F \right)}{\gamma m} + \sqrt{\frac{e^{-p}h}{m}} \right),$$

*where $\|.\|_{p,2}$ is the $\ell_p$ norm of the row $\ell_2$ norms.*

For $p$ of order $\ln h$, $\sqrt{\frac{h}{e^{-p}}} \approx$ constant improves on the $\sqrt{h}$ additive term in Theorem 2 and $h^{\frac{1}{2} - \frac{1}{p}} \left\| \mathbf{V}^T \right\|_{p,2} \approx h^{\frac{1}{2} - \frac{1}{\ln h}} \left\| \mathbf{V}^T \right\|_{\ln h, 2}$ which is a tight upper bound for $\| \mathbf{V} \|_F$ and is of the same order if all rows of $\mathbf{V}$ have the same norm - hence giving a tighter bound that decreases with $h$ for larger values. In particular for $p = \ln h$ we get the following bound.

**Corollary 6.** *Under the settings of Theorem 5, with probability $1 - \delta$ over the choice of the training set $\mathcal{S} = \{\mathbf{x}_i\}_{i=1}^m$, for any function $f(\mathbf{x}) = \mathbf{V}[\mathbf{U}\mathbf{x}]_+$, the generalization error is bounded as follows:*

$$L_0(f) \leq \hat{L}_\gamma(f) + \tilde{O}\left( \frac{\sqrt{c}h^{\frac{1}{2} - \frac{1}{\ln h}} \left\| \mathbf{V}^T \right\|_{\ln h, 2} \left( h^{\frac{1}{2} - \frac{1}{\ln h}} \left\| \mathbf{U} - \mathbf{U}^0 \right\|_{\ln h, 2} \left\| \mathbf{X} \right\|_F + \left\| \mathbf{U}_0\mathbf{X} \right\|_F \right)}{\gamma m} \right)$$

$$\leq \hat{L}_\gamma(f) + \tilde{O}\left( \frac{\sqrt{c}h^{\frac{1}{2} - \frac{1}{\ln h}} \left\| \mathbf{V}^T \right\|_{\ln h, 2} \left( h^{\frac{1}{2} - \frac{1}{\ln h}} \left\| \mathbf{U} - \mathbf{U}^0 \right\|_{\ln h, 2} + \left\| \mathbf{U}_0 \right\|_2 \right) \sqrt{\frac{1}{m} \sum_{i=1}^m \left\| \mathbf{x}_i \right\|_2^2}}{\gamma \sqrt{m}} \right).$$

# C  PROOFS

## C.1  PROOF OF THEOREM 1

We start by stating a simple lemma which is a vector-contraction inequality for Rademacher complexities and relates the norm of a vector to the expected magnitude of its inner product with a vector of Rademacher random variables. We use the following technical result from Maurer (2016) in our proof.

**Lemma 7** (Proposition 6 of Maurer (2016)). *Let $\xi_i$ be the Rademacher random variables. For any vector $\mathbf{v} \in \mathbb{R}^d$, the following holds:*

$$\|\mathbf{v}\|_2 \leq \sqrt{2} \mathop{\mathbb{E}}_{\xi_i \sim \{\pm 1\}, i \in [d]} [|\langle \boldsymbol{\xi}, \mathbf{v} \rangle|].$$

The above lemma can be useful to get Rademacher complexities in multi-class settings. The below lemma bounds the Rademacher-like complexity term for linear operators with multiple output centered around a reference matrix. The proof is very simple and similar to that of linear separators. See Bartlett & Mendelson (2002) for similar arguments.

**Lemma 8.** *For any positive integer c, positive scaler $r > 0$, reference matrix $\mathbf{V}^0 \in \mathbb{R}^{c \times d}$ and set $\{\mathbf{x}_i\}_{i=1}^m \subset \mathbb{R}^d$, the following inequality holds:*

$$\mathop{\mathbb{E}}_{\boldsymbol{\xi}_i \in \{\pm 1\}^c, i \in [m]} \left[ \sup_{\|\mathbf{V} - \mathbf{V}^0\|_F \leq r} \sum_{i=1}^m \langle \boldsymbol{\xi}_i, \mathbf{V} \mathbf{x}_i \rangle \right] \leq r \sqrt{c} \|\mathbf{X}\|_F.$$

*Proof.*

$$\mathop{\mathbb{E}}_{\boldsymbol{\xi}_i \in \{\pm 1\}^c, i \in [m]} \left[ \sup_{\|\mathbf{V} - \mathbf{V}^0\|_F \leq r} \sum_{i=1}^m \langle \boldsymbol{\xi}_i, \mathbf{V} \mathbf{x}_i \rangle \right]$$

$$= \mathop{\mathbb{E}}_{\boldsymbol{\xi}_i \in \{\pm 1\}^c, i \in [m]} \left[ \sup_{\|\mathbf{V} - \mathbf{V}_0\|_F \leq r} \left\langle \mathbf{V}, \sum_{i=1}^m \boldsymbol{\xi}_i \mathbf{x}_i^\top \right\rangle \right]$$

$$= \mathop{\mathbb{E}}_{\boldsymbol{\xi}_i \in \{\pm 1\}^c, i \in [m]} \left[ \sup_{\|\mathbf{V} - \mathbf{V}_0\|_F \leq r} \left\langle \mathbf{V} - \mathbf{V}_0 + \mathbf{V}_0, \sum_{i=1}^m \boldsymbol{\xi}_i \mathbf{x}_i^\top \right\rangle \right]$$

$$= \mathop{\mathbb{E}}_{\boldsymbol{\xi}_i \in \{\pm 1\}^c, i \in [m]} \left[ \sup_{\|\mathbf{V} - \mathbf{V}_0\|_F \leq r} \left\langle \mathbf{V} - \mathbf{V}_0, \sum_{i=1}^m \boldsymbol{\xi}_i \mathbf{x}_i^\top \right\rangle \right] + \mathop{\mathbb{E}}_{\boldsymbol{\xi}_i \in \{\pm 1\}^c, i \in [m]} \left[ \sup_{\|\mathbf{V} - \mathbf{V}_0\|_F \leq r} \left\langle \mathbf{V}_0, \sum_{i=1}^m \boldsymbol{\xi}_i \mathbf{x}_i^\top \right\rangle \right]$$

$$= \mathop{\mathbb{E}}_{\boldsymbol{\xi}_i \in \{\pm 1\}^c, i \in [m]} \left[ \sup_{\|\mathbf{V} - \mathbf{V}_0\|_F \leq r} \left\langle \mathbf{V} - \mathbf{V}_0, \sum_{i=1}^m \boldsymbol{\xi}_i \mathbf{x}_i^\top \right\rangle \right] + \mathop{\mathbb{E}}_{\boldsymbol{\xi}_i \in \{\pm 1\}^c, i \in [m]} \left[ \left\langle \mathbf{V}_0, \sum_{i=1}^m \boldsymbol{\xi}_i \mathbf{x}_i^\top \right\rangle \right]$$

$$= \mathop{\mathbb{E}}_{\boldsymbol{\xi}_i \in \{\pm 1\}^c, i \in [m]} \left[ \sup_{\|\mathbf{V} - \mathbf{V}_0\|_F \leq r} \left\langle \mathbf{V} - \mathbf{V}_0, \sum_{i=1}^m \boldsymbol{\xi}_i \mathbf{x}_i^\top \right\rangle \right]$$

$$\leq r \mathop{\mathbb{E}}_{\boldsymbol{\xi}_i \in \{\pm 1\}^c, i \in [m]} \left[ \left\| \sum_{i=1}^m \boldsymbol{\xi}_i \mathbf{x}_i^\top \right\|_F \right]$$

$$\overset{(i)}{\leq} r \mathop{\mathbb{E}}_{\boldsymbol{\xi}_i \in \{\pm 1\}^c, i \in [m]} \left[ \left\| \sum_{i=1}^m \boldsymbol{\xi}_i \mathbf{x}_i^\top \right\|_F^2 \right]^{1/2}$$

$$= r \left( \sum_{j=1}^c \mathop{\mathbb{E}}_{\xi \in \{\pm 1\}^m} \left[ \left\| \sum_{i=1}^m \xi_i \mathbf{x}_i^\top \right\|_F^2 \right] \right)^{1/2}$$

$$= r \sqrt{c} \|\mathbf{X}\|_F.$$

$(i)$ follows from the Jensen's inequality. □

We next show that the Rademacher complexity of the class of networks defined in (5) and (4) can be decomposed to that of hidden units.

**Lemma 9** (Rademacher Decomposition). *Given a training set $\mathcal{S} = \{\mathbf{x}_i\}_{i=1}^m$ and $\gamma > 0$, Rademacher complexity of the class $\mathcal{F}_\mathcal{W}$ defined in equations* (5) *and* (4) *is bounded as follows:*

$$\mathcal{R}_\mathcal{S}(\ell_\gamma \circ \mathcal{F}_\mathcal{W}) \leq \frac{\sqrt{2}}{\gamma m} \sum_{j=1}^h \mathop{\mathbb{E}}_{\boldsymbol{\xi}_i \in \{\pm 1\}^c, i \in [m]} \left[ \sup_{\|\mathbf{u}_j - \mathbf{u}_j^0\|_2 \leq \beta_j, \|\mathbf{v}_j\|_2 \leq \alpha_j} \sum_{i=1}^m \langle \boldsymbol{\xi}_i, \mathbf{v}_j \rangle \left[ \langle \mathbf{u}_j, \mathbf{x}_i \rangle \right]_+ \right]$$

*Proof.* We prove the inequality in the lemma statement using induction on $t$ in the following inequality:

$$m\mathcal{R}_\mathcal{S}(\ell_\gamma \circ \mathcal{F}_\mathcal{W}) \leq \mathop{\mathbb{E}}_{\boldsymbol{\xi}_i \in \{\pm 1\}^c, i \in [m]} \left[ \sup_{(\mathbf{V}, \mathbf{U}) \in \mathcal{W}} \frac{2}{\gamma} \sum_{i=1}^{t-1} \sum_{j=1}^h \langle \boldsymbol{\xi}_i, \mathbf{v}_j \rangle \left[ \langle \mathbf{u}_j, \mathbf{x}_i \rangle \right]_+ + \sum_{i=t}^m \xi_{i1} \ell_\gamma(\mathbf{V}[\mathbf{U}\mathbf{x}_i]_+, y_i) \right]$$

$$= \mathop{\mathbb{E}}_{\boldsymbol{\xi}_i \in \{\pm 1\}^c, i \in [m]} \left[ \sup_{(\mathbf{V}, \mathbf{U}) \in \mathcal{W}} \xi_{t1} \ell_\gamma(\mathbf{V}[\mathbf{U}\mathbf{x}_t]_+, y_t) + \phi^t_{\mathbf{V}, \mathbf{U}} \right],$$

where for simplicity of the notation, we let $\phi^t_{\mathbf{V}, \mathbf{U}} = \frac{\sqrt{2}}{\gamma} \sum_{i=1}^{t-1} \sum_{j=1}^h \langle \boldsymbol{\xi}_i, \mathbf{v}_j \rangle \langle \mathbf{u}_j, \mathbf{x}_i \rangle + \sum_{i=t+1}^m \xi_{i1} \ell_\gamma(\mathbf{V}[\mathbf{U}\mathbf{x}_i]_+, y_i)$. The above statement holds trivially for the base case of $t = 1$ by the definition of the Rademacher complexity (3). We now assume that it is true for any $t' \leq t$ and prove it is true for $t' = t + 1$.

$$m\mathcal{R}_\mathcal{S}(\ell_\gamma \circ \mathcal{F}_\mathcal{W}) \leq \mathop{\mathbb{E}}_{\boldsymbol{\xi}_i \in \{\pm 1\}^c, i \in [m]} \left[ \sup_{(\mathbf{V}, \mathbf{U}) \in \mathcal{W}} \xi_{t1} \ell_\gamma(\mathbf{V}[\mathbf{U}\mathbf{x}_t]_+, y_t) + \phi^t_{\mathbf{V}, \mathbf{U}} \right]$$

$$= \frac{1}{2} \mathop{\mathbb{E}}_{\boldsymbol{\xi}_i \in \{\pm 1\}^c, i \in [m]} \left[ \sup_{(\mathbf{V}, \mathbf{U}), (\mathbf{V}', \mathbf{U}') \in \mathcal{W}} \ell_\gamma(\mathbf{V}[\mathbf{U}\mathbf{x}_t]_+, y_t) - \ell_\gamma(\mathbf{V}'[\mathbf{U}'\mathbf{x}_t]_+, y_t) + \phi^t_{\mathbf{V}, \mathbf{U}} + \phi^t_{\mathbf{V}', \mathbf{U}'} \right]$$

$$\leq \frac{1}{2} \mathop{\mathbb{E}}_{\boldsymbol{\xi}_i \in \{\pm 1\}^c, i \in [m]} \left[ \sup_{(\mathbf{V}, \mathbf{U}), (\mathbf{V}', \mathbf{U}') \in \mathcal{W}} \frac{\sqrt{2}}{\gamma} \|\mathbf{V}[\mathbf{U}\mathbf{x}_t]_+ - \mathbf{V}'[\mathbf{U}'\mathbf{x}_t]_+\|_2 + \phi^t_{\mathbf{V}, \mathbf{U}} + \phi^t_{\mathbf{V}', \mathbf{U}'} \right]. \quad (9)$$

The last inequality follows from the $\frac{\sqrt{2}}{\gamma}$ Lipschitzness of the ramp loss. The ramp loss is $1/\gamma$ Lipschitz with respect to each dimension but since the loss at each point only depends on score of the correct labels and the maximum score among other labels, it is $\frac{\sqrt{2}}{\gamma}$-Lipschitz. By Lemma 7, the right

hand side of the above inequality can be bounded as follows:

$$
m\mathcal{R}_\mathcal{S}(\ell_\gamma \circ \mathcal{F}_\mathcal{W}) \le \mathop{\mathbb{E}}_{\boldsymbol{\xi}_i \in \{\pm 1\}^c, i \in [m]} \left[ \sup_{(\mathbf{V},\mathbf{U}) \in \mathcal{W}} \xi_{t1} \ell_\gamma(\mathbf{V}[\mathbf{U}\mathbf{x}_t]_+, y_t) + \phi^t_{\mathbf{V},\mathbf{U}} \right]
$$

$$
\le \frac{1}{2} \mathop{\mathbb{E}}_{\boldsymbol{\xi}_i \in \{\pm 1\}^c, i \in [m]} \left[ \sup_{(\mathbf{V},\mathbf{U}),(\mathbf{V}',\mathbf{U}') \in \mathcal{W}} \frac{\sqrt{2}}{\gamma} \|\mathbf{V}[\mathbf{U}\mathbf{x}_t]_+ - \mathbf{V}'[\mathbf{U}'\mathbf{x}_t]_+\|_2 + \phi^t_{\mathbf{V},\mathbf{U}} + \phi^t_{\mathbf{V}',\mathbf{U}'} \right]
$$

$$
\le \frac{1}{2} \mathop{\mathbb{E}}_{\boldsymbol{\xi}_i \in \{\pm 1\}^c, i \in [m]} \left[ \sup_{(\mathbf{V},\mathbf{U}),(\mathbf{V}',\mathbf{U}') \in \mathcal{W}} \frac{2}{\gamma} \mathop{\mathbb{E}}_{\boldsymbol{\xi}'_t \in \{\pm 1\}^c} [|\langle \boldsymbol{\xi}'_t, \mathbf{V}[\mathbf{U}\mathbf{x}_t]_+ - \mathbf{V}'[\mathbf{U}'\mathbf{x}_t]_+\rangle|] \right.
$$
$$
\left. + \phi^t_{\mathbf{V},\mathbf{U}} + \phi^t_{\mathbf{V}',\mathbf{U}'} \right]
$$

$$
\le \frac{1}{2} \mathop{\mathbb{E}}_{\boldsymbol{\xi}_t, \boldsymbol{\xi}_i \in \{\pm 1\}^c, i \in [m]} \left[ \sup_{(\mathbf{V},\mathbf{U}),(\mathbf{V}',\mathbf{U}') \in \mathcal{W}} \frac{2}{\gamma} |\langle \boldsymbol{\xi}'_t, \mathbf{V}[\mathbf{U}\mathbf{x}_t]_+ - \mathbf{V}'[\mathbf{U}'\mathbf{x}_t]_+\rangle| + \phi^t_{\mathbf{V},\mathbf{U}} + \phi^t_{\mathbf{V}',\mathbf{U}'} \right]
$$

$$
= \frac{1}{2} \mathop{\mathbb{E}}_{\boldsymbol{\xi}_i \in \{\pm 1\}^c, i \in [m]} \left[ \sup_{(\mathbf{V},\mathbf{U}),(\mathbf{V}',\mathbf{U}') \in \mathcal{W}} \frac{2}{\gamma} |\langle \boldsymbol{\xi}_t, \mathbf{V}[\mathbf{U}\mathbf{x}_t]_+ - \mathbf{V}'[\mathbf{U}'\mathbf{x}_t]_+\rangle| + \phi^t_{\mathbf{V},\mathbf{U}} + \phi^t_{\mathbf{V}',\mathbf{U}'} \right]
$$

$$
= \frac{1}{2} \mathop{\mathbb{E}}_{\boldsymbol{\xi}_i \in \{\pm 1\}^c, i \in [m]} \left[ \sup_{(\mathbf{V},\mathbf{U}),(\mathbf{V}',\mathbf{U}') \in \mathcal{W}} \frac{2}{\gamma} \langle \boldsymbol{\xi}_t, \mathbf{V}[\mathbf{U}\mathbf{x}_t]_+ - \mathbf{V}'[\mathbf{U}'\mathbf{x}_t]_+\rangle + \phi^t_{\mathbf{V},\mathbf{U}} + \phi^t_{\mathbf{V}',\mathbf{U}'} \right]
$$

$$
= \frac{1}{2} \mathop{\mathbb{E}}_{\boldsymbol{\xi}_i \in \{\pm 1\}^c, i \in [m]} \left[ \sup_{(\mathbf{V},\mathbf{U}) \in \mathcal{W}} \frac{2}{\gamma} \langle \boldsymbol{\xi}_t, \mathbf{V}[\mathbf{U}\mathbf{x}_t]_+\rangle + \phi^t_{\mathbf{V},\mathbf{U}} \right]
$$

$$
= \frac{1}{2} \mathop{\mathbb{E}}_{\boldsymbol{\xi}_i \in \{\pm 1\}^c, i \in [m]} \left[ \sup_{(\mathbf{V},\mathbf{U}) \in \mathcal{W}} \frac{2}{\gamma} \sum_{j=1}^h \langle \boldsymbol{\xi}_t, \mathbf{v}_j\rangle [\langle \mathbf{u}_j, \mathbf{x}_t\rangle]_+ + \phi^t_{\mathbf{V},\mathbf{U}} \right]
$$

$$
= \mathop{\mathbb{E}}_{\boldsymbol{\xi}_i \in \{\pm 1\}^c, i \in [m]} \left[ \sup_{(\mathbf{V},\mathbf{U}) \in \mathcal{W}} \frac{2}{\gamma} \sum_{i=1}^t \sum_{j=1}^h \langle \boldsymbol{\xi}_i, \mathbf{v}_j\rangle [\langle \mathbf{u}_j, \mathbf{x}_i\rangle]_+ + \sum_{i=t+1}^m \xi_{i1} \ell_\gamma(\mathbf{V}[\mathbf{U}\mathbf{x}_i]_+, y_i) \right]
$$

This completes the induction proof. $\qquad\square$

**Lemma 10** (Ledoux-Talagrand contraction, Ledoux & Talagrand (1991)). *Let $f : \mathbb{R}_+ \to \mathbb{R}_+$ be convex and increasing. Let $\phi_i : \mathbb{R} \to \mathbb{R}$ satisfy $\phi_i(0) = 0$ and be $L$-Lipschitz. Let $\xi_i$ be independent Rademacher random variables. For any $T \subseteq \mathbb{R}^n$,*

$$
\mathop{\mathbb{E}}_{\boldsymbol{\xi} \in \{\pm 1\}^m} f\left( \frac{1}{2} \sup_{t \in T} \left| \sum_{i=1}^n \xi_i \phi_i(t_i) \right| \right) \le \mathop{\mathbb{E}}_{\boldsymbol{\xi} \in \{\pm 1\}^m} f\left( L \sup_{t \in T} \left| \sum_{i=1}^n \xi_i t_i \right| \right)
$$

The above lemma will be used in the following proof of Theorem 1.

*Proof of Theorem 1.* By Lemma 9, we have:

$$\mathbb{E}_{\boldsymbol{\xi}_i \in \{\pm 1\}^c, i \in [m]} \left[ \sup_{\|\mathbf{u}_j - \mathbf{u}_j^0\|_2 \leq \beta_j, \|\mathbf{v}_j\|_2 \leq \alpha_j} \sum_{i=1}^m \langle \boldsymbol{\xi}_i, \mathbf{v}_j \rangle \left[ \langle \mathbf{u}_j, \mathbf{x}_i \rangle \right]_+ \right]$$

$$= \alpha_j \mathbb{E}_{\boldsymbol{\xi}_i \in \{\pm 1\}^c, i \in [m]} \left[ \sup_{\|\mathbf{u}_j - \mathbf{u}_j^0\|_2 \leq \beta_j} \left\| \sum_{i=1}^m [\langle \mathbf{u}_j, \mathbf{x}_i \rangle]_+ \boldsymbol{\xi}_i \right\|_2 \right]$$

$$\leq \alpha_j \mathbb{E}_{\boldsymbol{\xi}_i \in \{\pm 1\}^c, i \in [m]} \left[ \sup_{\|\mathbf{u}_j - \mathbf{u}_j^0\|_2 \leq \beta_j} \mathbb{E}_{\boldsymbol{\epsilon} \in \{\pm 1\}^c} \sqrt{2} \left| \left\langle \sum_{i=1}^m [\langle \mathbf{u}_j, \mathbf{x}_i \rangle]_+ \boldsymbol{\xi}_i, \boldsymbol{\epsilon} \right\rangle \right| \right]$$

$$\leq \sqrt{2} \alpha_j \mathbb{E}_{\boldsymbol{\epsilon} \in \{\pm 1\}^c} \mathbb{E}_{\boldsymbol{\xi}_i \in \{\pm 1\}^c, i \in [m]} \left[ \sup_{\|\mathbf{u}_j - \mathbf{u}_j^0\|_2 \leq \beta_j} \left| \left\langle \sum_{i=1}^m [\langle \mathbf{u}_j, \mathbf{x}_i \rangle]_+ \boldsymbol{\xi}_i, \boldsymbol{\epsilon} \right\rangle \right| \right]$$

$$= \sqrt{2} \alpha_j \mathbb{E}_{\boldsymbol{\xi}_i \in \{\pm 1\}^c, i \in [m]} \left[ \sup_{\|\mathbf{u}_j - \mathbf{u}_j^0\|_2 \leq \beta_j} \left| \left\langle \sum_{i=1}^m [\langle \mathbf{u}_j, \mathbf{x}_i \rangle]_+ \boldsymbol{\xi}_i, \mathbf{1} \right\rangle \right| \right]$$

$$= \sqrt{2} \alpha_j \mathbb{E}_{\boldsymbol{\xi}_i \in \{\pm 1\}^c, i \in [m]} \left[ \sup_{\|\mathbf{u}_j - \mathbf{u}_j^0\|_2 \leq \beta_j} \left| \sum_{i=1}^m \sum_{k=1}^c [\langle \mathbf{u}_j, \mathbf{x}_i \rangle]_+ \xi_{ik} \right| \right]$$

Now we can apply Lemma 10 with $n = m \times c$, $f(x) = x$, $T = \{(t_{ik})_{i,k=1,1}^{m,c} \mid t_{ik} = \langle \mathbf{u}_j, \mathbf{x}_i \rangle\}$, $\phi_{ik}(x) = [x]_+, \forall i \in [m], k \in [c]$, and we get

$$\sqrt{2} \alpha_j \mathbb{E}_{\boldsymbol{\xi}_i \in \{\pm 1\}^c, i \in [m]} \left[ \sup_{\|\mathbf{u}_j - \mathbf{u}_j^0\|_2 \leq \beta_j} \left| \sum_{i=1}^m \sum_{k=1}^c [\langle \mathbf{u}_j, \mathbf{x}_i \rangle]_+ \xi_{ik} \right| \right]$$

$$\leq 2\sqrt{2} \alpha_j \mathbb{E}_{\boldsymbol{\xi}_i \in \{\pm 1\}^c, i \in [m]} \left[ \sup_{\|\mathbf{u}_j - \mathbf{u}_j^0\|_2 \leq \beta_j} \left| \sum_{i=1}^m \sum_{k=1}^c \langle \mathbf{u}_j, \mathbf{x}_i \rangle \xi_{ik} \right| \right]$$

$$= 2\sqrt{2} \alpha_j \mathbb{E}_{\boldsymbol{\xi}_i \in \{\pm 1\}^c, i \in [m]} \left[ \sup_{\|\mathbf{u}_j - \mathbf{u}_j^0\|_2 \leq \beta_j} \sum_{i=1}^m \sum_{k=1}^c \langle \mathbf{u}_j, \mathbf{x}_i \rangle \xi_{ik} \right] \qquad (10)$$

$$\leq 2\sqrt{2} \alpha_j (\beta_j \|\mathbf{X}'\|_F + \|\mathbf{u}_j^0 \mathbf{X}'\|_2), \quad \text{where } \mathbf{X}' = [\mathbf{X}\, \mathbf{X} \, \dots \, \mathbf{X}] \in \mathbb{R}^{d \times cm}$$

$$\leq 2\sqrt{2c} \alpha_j (\beta_j \|\mathbf{X}\|_F + \|\mathbf{u}_j^0 \mathbf{X}\|_2)$$

The proof is completed by taking sum of above inequality over $j$ from 1 to $h$. $\qquad \square$

## C.2 PROOF OF THEOREMS 2 AND 5

We start by the following covering lemma which allows us to prove the generalization bound in Theorem 5 without assuming the knowledge of the norms of the network parameters. The following lemma shows how to cover an $\ell_p$ ball with a set that dominates the elements entry-wise, and bounds the size of a one such cover.

**Lemma 11** ($\ell_p$ covering lemma). *Given any $\epsilon, D, \beta > 0$, $p \geq 2$, consider the set $S_{p,\beta}^D = \{\mathbf{x} \in \mathbb{R}^D \mid \|\mathbf{x}\|_p \leq \beta\}$. Then there exist $N$ sets $\{T_i\}_{i=1}^N$ of the form $T_i = \{\mathbf{x} \in \mathbb{R}^D \mid |x_j| \leq \alpha_j^i, \forall j \in [D]\}$ such that $S_{p,\beta}^D \subseteq \bigcup_{i=1}^N T_i$ and $\|\boldsymbol{\alpha}^i\|_2 \leq D^{1/p-1/2} \beta(1 + \epsilon), \forall i \in [N]$ where $N = \binom{K+D-1}{D-1}$ and*

$$K = \left\lceil \frac{D}{(1+\epsilon)^p - 1} \right\rceil.$$

*Proof.* We prove the lemma by construction. Consider the set $Q = \left\{ \boldsymbol{\alpha} \in \mathbb{R}^D \mid \forall_i \alpha_i^p \in \{j\beta^p/K\}_{j=1}^K, \|\boldsymbol{\alpha}\|_p^p = \beta^p(1 + D/K) \right\}$. For any $\mathbf{x} \in S_{p,\beta}^D$, consider $\boldsymbol{\alpha}'$

such that for any $i \in [D]$, $\alpha_i' = \left( \left\lceil \frac{|x_i^p|K}{\beta^p} \right\rceil \frac{\beta^p}{K} \right)^{1/p}$. It is clear that $|x_i| \leq \alpha_i'$. Moreover, we have:

$$
\begin{aligned}
\|\boldsymbol{\alpha}'\|_p^p &= \sum_{i=1}^{D} \left\lceil \frac{|x_i^p|K}{\beta^p} \right\rceil \frac{\beta^p}{K} \\
&\leq \sum_{i=1}^{D} \left( \frac{|x_i^p|K}{\beta^p} + 1 \right) \frac{\beta^p}{K} \\
&= \|\mathbf{x}\|_p^p + \frac{D\beta^p}{K} \\
&\leq \beta^p \left( 1 + \frac{D}{K} \right)
\end{aligned}
$$

It is not possible to conclude $\boldsymbol{\alpha}' \in Q$ from the above inequality since $\|\boldsymbol{\alpha}\|_p^p$ is not necessarily equal to $\beta^p \left( 1 + \frac{D}{K} \right)$. However, it is possible to increase the magnitude of the last dimension of $\boldsymbol{\alpha}$ to grow its norm to the desired value. Let $\alpha_i'' = \alpha_i'$ for $i \in [D-1]$ and $\alpha_D'' = \left( \frac{\beta^p}{K} \left[ K + D - \left\lceil \frac{|x_i^p|K}{\beta^p} \right\rceil \right] \right)^{1/p}$. Therefore, we have $\boldsymbol{\alpha}'' \in Q$ and for any $i \in [D]$, $|x_i| \leq \alpha_i' \leq \alpha_i''$. Furthermore for any $\alpha \in Q$, we have:

$$
\begin{aligned}
\|\boldsymbol{\alpha}\|_2 &\leq D^{1/2-1/p} \|\boldsymbol{\alpha}\|_p \\
&= \beta^p D^{1/2-1/p}(1 + D/K) \\
&\leq \beta D^{1/2-1/p} \left( 1 + (1+\epsilon)^p - 1 \right)^{1/p} = \beta D^{1/2-1/p}(1+\epsilon)
\end{aligned}
$$

where the first inequality is based on relationship between $\ell_2$ and $\ell_p$ and the rest follows from definition of $Q$ and $K$. Therefore, to complete the proof, we only need to bound the size of the set $Q$. The size of the set $Q$ is equal to the number of unique solutions for the problem $\sum_{i=1}^{D} z_i = K + D$ for non-zero integer variables $z_i$, which is $\binom{K+D-1}{D-1}$. $\square$

**Lemma 12.** *For any $h, p \geq 2$, $d, c, \gamma, \mu > 0, \delta \in (0, 1)$ and $\mathbf{U}^0 \in \mathbb{R}^{h \times d}$, with probability $1 - \delta$ over the choice of the training set $\mathcal{S} = \{\mathbf{x}_i\}_{i=1}^{m} \subset \mathbb{R}^d$, for any function $f(\mathbf{x}) = \mathbf{V}[\mathbf{U}\mathbf{x}]_+$, such that $\mathbf{V} \in \mathbb{R}^{c \times h}, \mathbf{U} \in \mathbb{R}^{h \times d}, \|\mathbf{V}^\top\|_{p,2} \leq C_1, \|\mathbf{U} - \mathbf{U}^0\|_{p,2} \leq C_2$, the generalization error is bounded as follows:*

$$
\begin{aligned}
L_0(f) \leq \hat{L}_\gamma(f) &+ \frac{2\sqrt{2c}(\mu+1)^{\frac{2}{p}} h^{\frac{1}{2}-\frac{1}{p}} C_1 \left( h^{\frac{1}{2}-\frac{1}{p}} C_2 \|\mathbf{X}\|_F + \|\mathbf{U}^0\mathbf{X}\|_F \right)}{\gamma m} \\
&+ 3\sqrt{\frac{2\ln N_{p,h} + \ln(2/\delta)}{2m}},
\end{aligned}
$$

*where $N_{p,h} = \binom{\lceil h/\mu \rceil + h - 2}{h-1}$ and $\|.\|_{p,2}$ is the $\ell_p$ norm of the column $\ell_2$ norms.*

*Proof.* The proof of this lemma follows from using the result of Theorem 1 and taking a union bound to cover all the possible values of $\{\mathbf{V} \mid \|\mathbf{V}\|_{p,2} \leq C_1\}$ and $\mathbf{U} = \{\mathbf{U} \mid \|\mathbf{U} - \mathbf{U}^0\|_{p,2} \leq C_2\}$.

Note that $\forall \mathbf{x} \in \mathbb{R}^h$ and $p \geq 2$, we have $\|\mathbf{x}\|_2 \leq h^{\frac{1}{2}-\frac{1}{p}} \|\mathbf{x}\|_p$. Recall the result of Theorem 1, given any fixed $\boldsymbol{\alpha}, \boldsymbol{\beta}$, we have

$$
\begin{aligned}
\mathcal{R}_\mathcal{S}(\ell_\gamma \circ \mathcal{F}_\mathcal{W}) &\leq \frac{2\sqrt{2c}}{\gamma m} \|\boldsymbol{\alpha}\|_2 \left( \|\boldsymbol{\beta}\|_2 \|\mathbf{X}\|_F + \|\mathbf{U}^0\mathbf{X}\|_F \right) \\
&\leq \frac{2\sqrt{2c}}{\gamma m} h^{\frac{1}{2}-\frac{1}{p}} \|\boldsymbol{\alpha}\|_p \left( h^{\frac{1}{2}-\frac{1}{p}} \|\boldsymbol{\beta}\|_p \|\mathbf{X}\|_F + \|\mathbf{U}^0\mathbf{X}\|_F \right),
\end{aligned}
\tag{11}
$$

By Lemma 11, picking $\epsilon = ((1 + \mu)^{1/p} - 1)$, we can find a set of vectors, $\{\boldsymbol{\alpha}^i\}_{i=1}^{N_{p,h}}$, where $K = \left\lceil \frac{h}{\mu} \right\rceil$, $N_{p,h} = \binom{K+h-2}{h-1}$ such that $\forall \mathbf{x}, \|\mathbf{x}\|_p \leq C_1, \exists 1 \leq i \leq N_{p,h}, x_j \leq \alpha_j^i, \forall j \in [h]$. Similarly, picking $\epsilon = ((1 + \mu)^{1/p} - 1)$, we can find a set of vectors, $\{\boldsymbol{\beta}^i\}_{i=1}^{N_{p,h}}$, where $K = \left\lceil \frac{h}{\mu} \right\rceil$, $N_{p,h} = \binom{K+h-2}{h-1}$ such that $\forall \mathbf{x}, \|\mathbf{x}\|_p \leq C_2, \exists 1 \leq i \leq N_{p,h}, x_j \leq \beta_j^i, \forall j \in [h]$. $\qquad\square$

**Lemma 13.** *For any $h, p \geq 2$, $c, d, \gamma, \mu > 0$, $\delta \in (0, 1)$ and $\mathbf{U}^0 \in \mathbb{R}^{h \times d}$, with probability $1 - \delta$ over the choice of the training set $\mathcal{S} = \{\mathbf{x}_i\}_{i=1}^m \subset \mathbb{R}^d$, for any function $f(\mathbf{x}) = \mathbf{V}[\mathbf{U}\mathbf{x}]_+$ such that $\mathbf{V} \in \mathbb{R}^{c \times h}$ and $\mathbf{U} \in \mathbb{R}^{h \times d}$, the generalization error is bounded as follows:*

$$
L_0(f) \leq \hat{L}_\gamma(f) + \frac{4\sqrt{2c}(\mu + 1)^{\frac{2}{p}}(h^{\frac{1}{2} - \frac{1}{p}}\|\mathbf{V}^\top\|_{p,2} + 1)(h^{\frac{1}{2} - \frac{1}{p}}\|\mathbf{U} - \mathbf{U}_0\|_{p,2}\|\mathbf{X}\|_F + \|\mathbf{U}^0\mathbf{X}\|_F + 1)}{\gamma m}
$$
$$
+ 3\sqrt{\frac{\ln N_{p,h} + \ln(\gamma\sqrt{m}/\delta)}{m}},
$$

(12)

*where $N_{p,h} = \binom{\lceil h/\mu \rceil + h - 2}{h-1}$ and $\|.\|_{p,2}$ is the $\ell_p$ norm of the column $\ell_2$ norms.*

*Proof.* This lemma can be proved by directly applying union bound on Lemma 12 with for every $C_1 \in \left\{ \frac{i}{h^{1/2 - 1/p}} \mid i \in \left[ \left\lceil \frac{\gamma\sqrt{m}}{4} \right\rceil \right] \right\}$ and every $C_2 \in \left\{ \frac{i}{h^{1/2 - 1/p}\|\mathbf{X}\|_F} \mid i \in \left[ \left\lceil \frac{\gamma\sqrt{m}}{4} \right\rceil \right] \right\}$. For $\|\mathbf{V}^\top\|_{p,2} \leq \frac{1}{h^{1/2 - 1/p}}$, we can use the bound where $C_1 = 1$, and the additional constant 1 in Eq. 12 will cover that. The same is true for the case of $\|\mathbf{U}\|_{p,2} \leq \frac{i}{h^{1/2 - 1/p}\|\mathbf{X}\|_F}$. When any of $h^{1/2 - 1/p}\|\mathbf{V}^\top\|_{p,2}$ and $h^{1/2 - 1/p}\|\mathbf{X}\|_F\|\mathbf{U}\|_{p,2}$ is larger than $\left\lceil \frac{\gamma\sqrt{m}}{4} \right\rceil$, the second term in Eq. 12 is larger than 1 thus holds trivially. For the rest of the case, there exists $(C_1, C_2)$ such that $h^{1/2 - 1/p}C_1 \leq h^{1/2 - 1/p}\|\mathbf{V}^\top\|_{p,2} + 1$ and $h^{1/2 - 1/p}C_2 \leq h^{1/2 - 1/p}\|\mathbf{X}\|_F\|\mathbf{X}\|_F\|\mathbf{U}\|_{p,2} + 1$. Finally, we have $\frac{\gamma\sqrt{m}}{4} \geq 1$ otherwise the second term in Eq. 12 is larger than 1. Therefore, $\left\lceil \frac{\gamma\sqrt{m}}{4} \right\rceil \leq \frac{\gamma\sqrt{m}}{4} + 1 \leq \frac{\gamma\sqrt{m}}{2}$. $\qquad\square$

We next use the general results in Lemma 13 to give specific results for the case $p = 2$.

**Lemma 14.** *For any $h \geq 2$, $c, d, \gamma > 0$, $\delta \in (0, 1)$ and $\mathbf{U}^0 \in \mathbb{R}^{h \times d}$, with probability $1 - \delta$ over the choice of the training set $\mathcal{S} = \{\mathbf{x}_i\}_{i=1}^m \subset \mathbb{R}^d$, for any function $f(\mathbf{x}) = \mathbf{V}[\mathbf{U}\mathbf{x}]_+$ such that $\mathbf{V} \in \mathbb{R}^{c \times h}$ and $\mathbf{U} \in \mathbb{R}^{h \times d}$, the generalization error is bounded as follows:*

$$
L_0(f) \leq \hat{L}_\gamma(f) + \frac{6\sqrt{c}(\|\mathbf{V}^\top\|_F + 1)(\|\mathbf{U} - \mathbf{U}_0\|_F\|\mathbf{X}\|_F + \|\mathbf{U}^0\mathbf{X}\|_F + 1)}{\gamma m}
$$
$$
+ 3\sqrt{\frac{5h + \ln(\gamma\sqrt{m}/\delta)}{m}},
$$

(13)

*Proof.* To prove the lemma, we directly upper bound the generalization bound given in Lemma 13 for $p = 2$ and $\mu = \frac{3\sqrt{2}}{4} - 1$. For this choice of $\mu$ and $p$, we have $4(\mu + 1)^{2/p} \leq 3\sqrt{2}$ and $\ln N_{p,h}$ is bounded as follows:

$$
\ln N_{p,h} = \ln \binom{\lceil h/\mu \rceil + h - 2}{h-1} \leq \ln \left( \left[ e\frac{\lceil h/\mu \rceil + h - 2}{h-1} \right]^{h-1} \right) = (h-1)\ln \left( e + e\frac{\lceil h/\mu \rceil - 1}{h-1} \right)
$$
$$
\leq (h-1)\ln \left( e + e\frac{h/\mu}{h-1} \right) \leq h\ln(e + 2e/\mu) \leq 5h
$$

$\qquad\square$

*Proof of Theorem 2.* The proof directly follows from Lemma 14 and using $\tilde{O}$ notation to hide the constants and logarithmic factors. $\qquad\square$

Next lemma states a generalization bound for any $p \geq 2$, which is looser than 14 for $p = 2$ due to extra constants and logarithmic factors.

**Lemma 15.** *For any $h, p \geq 2$, $c, d, \gamma > 0$, $\delta \in (0,1)$ and $\mathbf{U}^0 \in \mathbb{R}^{h \times d}$, with probability $1 - \delta$ over the choice of the training set $\mathcal{S} = \{\mathbf{x}_i\}_{i=1}^m \subset \mathbb{R}^d$, for any function $f(\mathbf{x}) = \mathbf{V}[\mathbf{U}\mathbf{x}]_+$ such that $\mathbf{V} \in \mathbb{R}^{c \times h}$ and $\mathbf{U} \in \mathbb{R}^{h \times d}$, the generalization error is bounded as follows:*

$$L_0(f) \leq \hat{L}_\gamma(f) + \frac{4e^2\sqrt{2c}(h^{\frac{1}{2}-\frac{1}{p}}\|\mathbf{V}^\top\|_{p,2} + 1)\left(h^{\frac{1}{2}-\frac{1}{p}}\|\mathbf{U} - \mathbf{U}_0\|_{p,2}\|\mathbf{X}\|_F + \|\mathbf{U}^0\mathbf{X}\|_F + 1\right)}{\gamma m}$$
$$+ 3\sqrt{\frac{\lceil e^{1-p}h - 1\rceil \ln(eh) + \ln(\gamma\sqrt{m}/\delta)}{m}},$$

(14)

$\|.\|_{p,2}$ *is the $\ell_p$ norm of the column $\ell_2$ norms.*

*Proof.* To prove the lemma, we directly upper bound the generalization bound given in Lemma 13 for $\mu = e^p - 1$. For this choice of $\mu$ and $p$, we have $(\mu + 1)^{2/p} = e^2$. Furthermore, if $\mu \geq h$, $N_{p,h} = 0$, otherwise $\ln N_{p,h}$ is bounded as follows:

$$\ln N_{p,h} = \ln\binom{\lceil h/\mu\rceil + h - 2}{h - 1} = \ln\binom{\lceil h/\mu\rceil + h - 2}{\lceil h/\mu\rceil - 1} \leq \ln\left(\left[e\frac{\lceil h/\mu\rceil + h - 2}{\lceil h/\mu\rceil - 1}\right]^{\lceil h/\mu\rceil - 1}\right)$$

$$= (\lceil h/(e^p - 1)\rceil - 1)\ln\left(e + e\frac{h - 1}{\lceil h/(e^p - 1)\rceil - 1}\right) \leq (\lceil e^{1-p}h\rceil - 1)\ln(eh)$$

Since the right hand side of the above inequality is greater than zero for $\mu \geq h$, it is true for every $\mu > 0$. $\square$

*Proof of Theorem 5.* The proof directly follows from Lemma 15 and using $\tilde{O}$ notation to hide the constants and logarithmic factors. $\square$

### C.3 PROOF OF THE LOWER BOUND

*Proof of Theorem 3.* We will start with the case $h = d = 2^k$, $m = n2^k$ for some $k, n \in \mathbb{N}$.

We will pick $\mathbf{V} = \boldsymbol{\alpha}^\top = [\alpha_1 \ldots \alpha_{2^k}]$ for every $\boldsymbol{\xi}$, and $\mathcal{S} = \{\mathbf{x}_i\}_{i=1}^m$, where $\mathbf{x}_i := \mathbf{e}_{\lceil\frac{i}{n}\rceil}$. That is, the whole dataset are divides into $2^k$ groups, while each group has $n$ copies of a different element in standard orthonormal basis.

We further define $\epsilon_j(\boldsymbol{\xi}) = \sum_{(j-1)n+1}^{jn} \xi_i$, $\forall j \in [2^k]$ and $\mathbf{F} = (\boldsymbol{f}_1, \boldsymbol{f}_2, \ldots, \boldsymbol{f}_{2^k}) \in \{-2^{-k/2}, 2^{-k/2}\}^{2^k \times 2^k}$ be the Hadamard matrix which satisfies $\langle \boldsymbol{f}_i, \boldsymbol{f}_j\rangle = \delta_{ij}$. Note that for $\mathbf{s} \in \mathbb{R}^d$, $s_j = \alpha_j\beta_j$, $\forall j \in [d]$, it holds that

$$\forall i \in [d], \quad \max\{\langle\mathbf{s}, [\mathbf{f}_i]_+\rangle, \langle\mathbf{s}, [-\mathbf{f}_i]_+\rangle\} \geq \frac{1}{2}\left(\langle\mathbf{s}, [\mathbf{f}_i]_+\rangle + \langle\mathbf{s}, [-\mathbf{f}_i]_+\rangle\right) = \frac{\sum_{j=1}^{2^k} s_j|f_{ji}|}{2} = 2^{-\frac{k}{2}-1}\boldsymbol{\alpha}^\top\boldsymbol{\beta}.$$

Thus without loss of generality, we can assume $\forall i \in [2^k]$, $\langle\mathbf{s}, [\mathbf{f}_i]_+\rangle \geq 2^{-\frac{k}{2}-1}\boldsymbol{\alpha}^\top\boldsymbol{\beta}$ by flipping the signs of $\mathbf{f}_i$.

For any $\boldsymbol{\xi} \in \{-1, 1\}^n$, let $\text{Diag}(\boldsymbol{\beta})$ be the square diagonal matrix with its diagonal equal to $\boldsymbol{\beta}$ and $\widetilde{\mathbf{F}}(\boldsymbol{\xi})$ be the following:

$$\widetilde{\mathbf{F}}(\boldsymbol{\xi}) := [\widetilde{\mathbf{f}}_1, \widetilde{\mathbf{f}}_2, \ldots, \widetilde{\mathbf{f}}_{2^k}] \text{ such that if } \epsilon_i(\boldsymbol{\xi}) \geq 0, \widetilde{\mathbf{f}}_i = \boldsymbol{f}_i, \text{ and if } \epsilon_i(\boldsymbol{\xi}) < 0, \widetilde{\mathbf{f}}_i = \mathbf{0},$$

and we will choose $\mathbf{U}(\boldsymbol{\xi})$ as $\text{Diag}(\boldsymbol{\beta}) \times \widetilde{\mathbf{F}}(\boldsymbol{\xi})$.

Since $\mathbf{F}$ is orthonormal, by the definition of $\widetilde{\mathbf{F}}(\boldsymbol{\xi})$, we have $\|\widetilde{\mathbf{F}}(\boldsymbol{\xi})\|_2 \leq 1$ and the 2-norm of each row of $\widetilde{\mathbf{F}}$ is upper bounded by 1. Therefore, we have $\|\mathbf{U}(\boldsymbol{\xi})\|_2 \leq \|\text{Diag}(\boldsymbol{\beta})\|_2\|\widetilde{\mathbf{F}}(\boldsymbol{\xi})\|_2 \leq \max_i \beta_i$, and

$\|\mathbf{u}_i - \mathbf{u}_i^0\|_2 = \|\mathbf{u}^i\|_2 \leq \beta_i \|\mathbf{e}_i^\top \mathbf{F}(\widetilde{\boldsymbol{\xi}})\|_2 \leq \beta_i$. In other words, $f(\mathbf{x}) = \mathbf{V}[\mathbf{U}(\boldsymbol{\xi})\mathbf{x}]_+ \in \mathcal{F}_{\mathcal{W}'}$. We will omit the index $\boldsymbol{\xi}$ when it's clear.

Now we have

$$\sum_{i=1}^n \xi_i \mathbf{V}[\mathbf{U}\mathbf{x}_i]_+ = \sum_{j=1}^{2^k} \sum_{i=(j-1)n+1}^{jn} \xi_i \mathbf{V}[\mathbf{U}\mathbf{x}_i]_+ = \sum_{j=1}^{2^k} \epsilon_j(\boldsymbol{\xi}) \mathbf{V}[\mathbf{U}\mathbf{e}_j]_+.$$

Note that

$$\epsilon_j \mathbf{V}[\mathbf{U}\mathbf{e}_j]_+ = \epsilon_j \left\langle \text{Diag}(\boldsymbol{\beta})\boldsymbol{\alpha}, [\widetilde{\mathbf{f}}_j]_+ \right\rangle = \epsilon_j \left\langle \mathbf{s}, [\mathbf{f}_j]_+ \right\rangle \mathbf{1}_{\epsilon_j > 0} \geq 2^{-\frac{k}{2}-1} \boldsymbol{\alpha}^\top \boldsymbol{\beta} [\epsilon_j]_+.$$

The last inequality uses the previous assumption, that $\forall j \in [2^k]$, $\langle \mathbf{s}, [\mathbf{f}_j]_+ \rangle \geq 2^{-\frac{k}{2}-1} \boldsymbol{\alpha}^\top \boldsymbol{\beta}$.

Thus,

$$\begin{aligned}
m\mathcal{R}_\mathcal{S}(\mathcal{F}_{\mathcal{W}'}) &\geq \mathop{\mathbb{E}}_{\boldsymbol{\xi} \sim \{\pm 1\}^m} \left[ \sum_{i=1}^m \xi_i \mathbf{V}[\mathbf{U}(\boldsymbol{\xi})\mathbf{x}_i]_+ \right] \\
&\geq \boldsymbol{\alpha}^\top \boldsymbol{\beta} 2^{-\frac{k}{2}-1} \mathop{\mathbb{E}}_{\boldsymbol{\xi} \sim \{\pm 1\}^m} \left[ \sum_{j=1}^{2^k} [\epsilon_j(\boldsymbol{\xi})]_+ \right] \\
&= \boldsymbol{\alpha}^\top \boldsymbol{\beta} 2^{\frac{k}{2}-1} \mathop{\mathbb{E}}_{\boldsymbol{\xi} \sim \{\pm 1\}^n} [[\epsilon_1(\boldsymbol{\xi})]_+] \\
&= \boldsymbol{\alpha}^\top \boldsymbol{\beta} 2^{\frac{k}{2}-2} \mathop{\mathbb{E}}_{\boldsymbol{\xi} \sim \{\pm 1\}^n} [|\epsilon_1(\boldsymbol{\xi})|] \\
&\geq \boldsymbol{\alpha}^\top \boldsymbol{\beta} 2^{\frac{k-1}{2}-2} \sqrt{n} \\
&= \frac{\boldsymbol{\alpha}^\top \boldsymbol{\beta} \sqrt{2d}}{8} \sqrt{\frac{m}{d}} \\
&= \frac{\boldsymbol{\alpha}^\top \boldsymbol{\beta} \sqrt{2m}}{8}
\end{aligned}$$

where the last inequality is by Lemma 7.

For arbitrary $d = h \leq m$, $d, h, m \in \mathbb{Z}_+$, let $k = \lfloor \log_2 d \rfloor$, $d' = h' = 2^k$, $m' = \lfloor \frac{m}{2^k} \rfloor * 2^k$. Then we have $h' \geq \frac{h}{2}$, $m' \geq \frac{m}{2}$. Thus there exists $S \subseteq [h]$, such that $\sum_{i \in S} \alpha_i \beta_i \geq \sum_{i=1}^h \alpha_i \beta_i$. Therefore we can pick $h'$ hidden units out of $h$ hidden units, $d'$ input dimensions out of $d$ dimensions, $m'$ input samples out of $m$ to construct a lower bound of $\frac{\sum_{i \in S} \alpha_i \beta_i \sqrt{2m'}}{8} \geq \frac{\boldsymbol{\alpha}^\top \boldsymbol{\beta} \sqrt{m}}{16}$. $\qquad\square$

