# OpenReview forum: "The role of over-parametrization in generalization of neural networks"
_ICLR.cc/2019/Conference_

### Official Review · AnonReviewer3 · 2018-11-02
**The authors present a novel bound for the generalization error of 1-layer neural networks with multiple outputs and ReLU activations.**

**Rating:** 7
**Confidence:** 3

**Review:**

It is shown empirically that common algorithms used in supervised learning (SGD) yield networks for which such upper bound decreases as the number of hidden units increases. This might explain why in some cases overparametrized models have better generalization properties.

This paper tackles the important question of why in the context of supervised learning, overparametrized neural networks in practice generalize better. First, the concepts of \textit{capacity} and \textit{impact} of a hidden unit are introduced. Then, {\bf Theorem 1} provides an upper bound for the empirical Rademacher complexity of the class of 1-layer networks with hidden units of bounded \textit{capacity} and \textit{impact}. Next, {\bf Theorem 2} which is the main result, presents a new upper bound for the generalization error of 1-layer networks. An empirical comparison with existing generalization bounds is made and the presented bound is the only one that in practice decreases when the number of hidden units grows. Finally {\bf Theorem 3} is presented, which provides a lower bound for the Rademacher complexity of a class of neural networks, and such bound is compared with existing lower bounds.

## Strengths
- The paper is theoretically sound, the statement of the theorems
    are clear and the authors seem knowledgeable when bounding the
    generalization error via Rademacher complexity estimation.

- The paper is readable and the notation is consistent throughout.

- The experimental section is well described, provides enough empirical
    evidence for the claims made, and the plots are readable and well
    presented, although they are best viewed on a screen.

- The appendix provides proofs for the theoretical claims in the
    paper. However, I cannot certify that they are correct.

- The problem studied is not new, but to my knowledge the
    presented bounds are novel and the concepts of capacity and
    impact are new. Theorem 3 improves substantially over
    previous results.

- The ideas presented in the paper might be useful for other researchers
    that could build upon them, and attempt to extend and generalize
    the results to different network architectures.

- The authors acknowledge that there might be other reasons
    that could also explain the better generalization properties in the
    over-parameterized regime, and tone down their claims accordingly.

## Weaknesses
\begin{itemize}
- The abstract reads "Our capacity bound correlates with the behavior
    of test error with increasing network sizes ...", it should
    be pointed out that the actual bound increases with increasing
    network size (because of a sqrt(h/m) term), and that such claim
    holds only in practice.

- In page 8 (discussion following Theorem 3) the claim
    "... all the previous capacity lower bounds for spectral
        norm bounded classes of neural networks (...) correspond to
        the Lipschitz constant of the network. Our lower bound strictly
    improves over this ...", is not clear. Perhaps a more concise
    presentation of the argument is needed. In particular it is not clear
    how a lower bound for the Rademacher complexity of F_W translates into a
    lower bound for the rademacher complexity of l_\gamma F_W. This makes the claim of tightness of Theorem 1 not clear. Also this makes
    the initial claim about the tightness of Theorem 2 not clear.

---

> ### Author Response · Authors · 2018-11-15
> **Thanks for your positive feedback and suggested improvements.**
>
> Thanks for your positive feedback and suggested improvements.
>
> 1) We have not claimed in the paper that our bound decreases with the network size but rather shows correlation with the test error, which is an empirical observation. To make this very clear, we have updated the abstract to emphasize that the correlation of the bound with the test error is for network sizes within the range reported in the experiments.
>
> 2) Since the l_gamma loss is (\sqrt{2}/gamma)-Lipschitz, the Rademacher complexity of l_gamma o F is (\sqrt{2}/gamma) times Rademacher complexity of F so the important object to calculate the complexity measure is F and our lower bound is given for F. We will clarify this confusion in the final version.

---

### Official Review · AnonReviewer2 · 2018-11-12
**Promising paper, with a couple of clarifications needed**

**Rating:** 7
**Confidence:** 5

**Review:**

Let me start by apologizing for the delayed review - in fact I was asked today to replace an earlier assigned reviewer. Hopefully the clarifications I request won't be too time consuming to meet the deadline coming up.

###

First of all, the problem which the authors are attempting to answer is quite important: the effect of over-parametrization is not well understood on a theoretical level. As the paper illustrate, 2-layer networks are already capable of generalizing while being over-parameterized, therefore justifying their setting.

Next this paper motivates the study of complexity quantities that tend to decrease with the number of parameters, in particular figure 3 motivates the conjecture that the complexity measure in Theorem 2 can control generalization error. The paper also does a great job comparing related work, motivating their results.

###

At this point, I would like to request a couple of clarifications in the proofs. Perhaps it's due to the fact that I only spent a day reading, but at least I think we could improve on its readability. Regardless, I currently do not yet trust a couple of the proofs, and I believe the acceptance of this paper should be conditioned on confirming the correctness of these proofs.

(1) Let's start with Lemma 10. In the middle equation block, we obtain a bound
  \| alpha^prime \|_p^p <= beta^p ( 1 + D/K )
and the proof concludes alpha^prime is in Q. However this cannot be the case for all alpha^prime.

Consider x=0 which is in S_{p, beta}^D, then we have alpha^prime = 0 as well. In the definition of Q, we require all the j's to sum up to K+D, which is not met here.

At the same time, the next claim
  \| alpha \|_2 <= D^{1/2 - 1/p} \| alpha^prime \|_p
does not seem to follow from the above calculations. In particular, alpha^prime seems to be defined with respect to an x in S_{p, beta}, however in this case we did not specify such an x. Perhaps did you mean there exist such an alpha^prime?

(2) In the proof of Theorem 3, there is an important inequality needed to complete the proof
  max{ <s, f_i> , <s, -f_i> } >= 1/2 * ( <s, [f_i]_+> + <s, [-f_i]_+> )

Perhaps I am missing something obvious, but I believe this inequality fails when we choose s as a constant vector, and f_i to have the same number of positive and negative signs (which is possible in a Hadamard matrix). In this case, the left hand side should be equal to zero, where as the right hand side will be positive.

###

To summarize, if these proofs can be confirmed, I believe this paper would have made significant contribution to the problem of over-parametrization in deep learning, and of course should be accepted.

###

I corrected several typos and found minor issues as I read, perhaps this will be useful to improve readability as well.

Page 13, proof of Lemma 8
  - after the V_0 term is separated, there is a sup over \|V_0\|_F <= r in the expectation, which should be \|V-V_0\|_F <= r instead.

Page 14, Lemma 9
  - the lemma did not define rho_{ij} in the statement

Page 15, proof of Lemma 9
  - in equation (12), there is an x_y vector that should x_t

Page 15, proof of Theorem 1
  - while I eventually figured it out, it's unclear how Lemma 8 is applied here. Perhaps one more step identifying the exact matrices in the statement of Lemma 8 will be helpful to future readers, and maybe explain where the sqrt(2) factor come from as well.

Page 16, proof of Lemma 10
  - in the beginning of the proof, to stay consistent with the notation, we should replace S_{p, beta} with S_{p, beta}^D
  - I believe the cardinality of Q should be (K + D - 1) choose (D - 1), as we need to choose positive j's to sum up to (K+D) in the definition of Q. This reduces down to the problem of choosing natural numbers j's summing K, which is (K+D-1) choose (D-1). Consider the stack exchange post here:
https://math.stackexchange.com/questions/919676/the-number-of-integer-solutions-of-equations

Page 16, proof and statement of Lemma 11
  - I believe in the first term, the factor should be m instead of sqrt(m). I think the mistake happened when applying the union bound, as it should only affect the term containing delta

Page 17, Lemma 12
  - same as Lemma 11, we should have m instead of sqrt(m)

Page 18, proof of Theorem 3
  - at the bottom the statement "F is orthogonal" does not imply the norm is less than 1, but rather we should say "F is orthonormal"

Page 19, proof of Theorem 3
  - at the top, "we will omit the index epsilon" should be "xi" instead
  - in the final equation block, we have the Rademacher complexity of F_{W_2}, instead it should be F_{W^prime}

---

> ### Author Response · Authors · 2018-11-15
> **Clarification of the Proofs**
>
> Thanks a lot for reading our paper very carefully and helping us improve the readability and validity of the proofs with your suggestions. We are glad that you found our paper to be a significant contribution to the understanding of over-parameterization in deep learning. We have applied all your suggestions in the revision which is uploaded in the openreview. Here we clarify the two issues you raised regarding the proofs:
>
> 1) Lemma 10: As you guessed, it is indeed the case that the precise way to state is that “there exist such \alpha’’ “. This \alpha’’ can be constructed by simply increasing the value along the last dimension of the \alpha’ to get the desired norm. We have updated the paper with the clarification.
>
> 2) Theorem 3: You are right about the inequality in the proof of Theorem 3. This was a typo which can be fixed by replacing max{ <s, f_i> , <s, -f_i> } by max{ <s, [f_i]_+> , <s, [-f_i]_+> } in the left hand side. And this is indeed the quantity we use in the later part of the proof. We have corrected this typo in the revision.
>
> Given that we have resolved the two issues you raised, we respectfully ask you to increase the score to reflect the significance of this work on understanding the role of over-parameterization in neural networks. We thank you again for your valuable feedback.

---

> ### Comment · AnonReviewer2 · 2018-11-15
> **All majors issues fixed**
>
> Thank you for the quick reply, at this point I believe both of the major issues are properly addressed, and the proofs are rigorous. As promised, I would recommend accepting this paper.
>
> One more minor typo in Lemma 10 - in the last equation block where we plug in the value of \| alpha \|_p, I believe you initially plugged in the value of p-th power of it. Instead I believe it should be
>   beta D^{1/2 - 1/p} (1 + D/K)^{1/p}
> Once again, this is a very minor issue, and I can see the rest of the results follow from this correction.

---

### Official Review · AnonReviewer4 · 2018-11-20
**Solid paper.**

**Rating:** 7
**Confidence:** 3

**Review:**

The authors aim to shed light on the role of over-parametrization in generalization error. They do so for the special case of 2 layer fully connected ReLU networks, a "simple" setting where one still sees empirically that the test error decreasing as over-parametrization increases.

Based on empirical observations of norms (and norms relative to initialization) in trained overparametrized networks, the authors are led to the definition of a new norm-bounded class of neural networks. Write u_i for the vector of weights incoming to hidden node i. Write v_i for the weights outgoing from hidden node i. They study classes where the Euclidean norm of v_i is bounded by a constant alpha_i and where the Euclidean norm of u_i - u^0_i is bounded by beta_i, where u^0_i is the value of u_i after random initialization. Call this class F_{alpha,beta} where alpha,beta are specific vectors of bounds.

The main result is a bound on the empirical Rademacher complexity of F_{alpha,beta}.
The authors also given lower bounds on the empirical Rademacher complexity for carefully chosen data points, showing that the bounds are tight. These Rademacher bounds yield standard bounds on the ramp loss for fixed alpha,beta, and margin, and then a union bound argument extends the bound to data-dependent alpha,beta and margin.

The authors compare the bounds to existing norm-based bounds in the literature. The basic argument is that the terms in other bounds tend to grow as networks get much larger, while their terms shrink. Note that at no point are the bounds in this paper "nonvacuous", ie they are always larger than one.

In summary, I think this is a strong paper. The explanatory power of the results are still oversold in my opinion, even if they use hedged language like "could explain the role...". But the work is definitely pointing the way towards an explanation and deserves publication. The technical results in the appendix will be of interest to the learning theory community.

issues:

"could explain role of over-parametrization". Perhaps this work might point the way to an explanation, but it does not yet provide an explanation.  It is a big improvement it seems.

"bound improves over the existing bounds". From this statement and the discussion comparing the bounds, it is not clear whether this bound formally dominates existing bounds or merely does so empirically (or under empirical conditions).

typos:

bigger than the Lipschitz CONSTANT of the network class

H undefined

Rademacher defined for H but must be defined on loss class (or a generic function class, not H)

"we need to cover" --> "it suffices to"

"the following two inequaliTIES hold by Lemma 8"

bibliography is a mess: half of the arxiv papers are published. typos everywhere, very sloppy.

(This review was requested late in the process due to another reviewer dropping out of the process.)

[UPDATE]. The authors addressed my concerns stated in my review above. I think the bibliography has improved and I recommend acceptance.

---

> ### Author Response · Authors · 2018-11-25
> **Thanks for your feedback - All comments are addressed in the revision**
>
> Thank you for your valuable feedback. We have uploaded a revision addressing all your comments.
>
> In particular, we have made the following changes:
>
> 1) Improved the bibliography significantly
> 2) Toned down the claims of the paper
> 3) Fixed typos.
>
> Thanks again for pointing to these issues.

---

### Author Response · Authors · 2018-11-25
**Final revision is uploaded - All reviewers' comments are addressed - Thank you for your valuable feedback**

We thank all reviewers for their useful feedback. The final revision is uploaded . This version has addressed all reviewers' comments. We believe that the quality of our paper has improved in the discussion process. We again thank all reviewers for their time and effort.

---

### Meta-Review · Area_Chair1 · 2018-12-16
**strong paper**

**Confidence:** 4
**Recommendation:** Accept (Poster)

**Metareview:**

I agree with the reviewers that this is a strong contribution and provides new insights, even if it doesn't quite close the problem.

p.s.: It seems that centering the weight matrices at initialization is a key idea. The authors note that Dziugaite and Roy used  bounds that were based on the distance to initialization, but that their reported numerical generalization bounds also increase with the increasing network size. Looking back at that work, they look at networks where the size increases by a very large factor (going from e.g. 400,000 parameters roughly to over 1.2 million, so a factor of 2.5), at the same time the bound increases by a much smaller factor. The type of increase also seems much less severe than those pictured in Figures 3/5. Since Dzugate and Roy's bounds involved optimization, perhaps the increase there is merely apparent.